# PPARɣ drives IL-33-dependent ILC2 pro-tumoral functions

Giuseppe Ercolano [1,2], Alejandra Gomez-Cadena[1,2], Nina Dumauthioz[3], Giulia Vanoni[3], Mario Kreutzfeldt[4], Tania Wyss[3], Liliane Michalik[5], Romain Loyon[6,7], Angela Ianaro[8], Ping-Chih Ho [3], Christophe Borg[6,7], Manfred Kopf [9], Doron Merkler [4], Philippe Krebs [10], Pedro Romero [3], Sara Trabanelli[1,2,11] & Camilla Jandus [1,2,11 ✉]

Group 2 innate lymphoid cells (ILC2s) play a critical role in protection against helminths and in diverse inflammatory diseases by responding to soluble factors such as the alarmin IL-33, that is often overexpressed in cancer. Nonetheless, regulatory factors that dictate ILC2 functions remain poorly studied. Here, we show that peroxisome proliferator-activated receptor gamma (PPARγ) is selectively expressed in ILC2s in humans and in mice, acting as a central functional regulator. Pharmacologic inhibition or genetic deletion of PPARγ in ILC2s significantly impair IL-33-induced Type-2 cytokine production and mitochondrial fitness. Further, PPARγ blockade in ILC2s disrupts their pro-tumoral effect induced by IL-33-secreting cancer cells. Lastly, genetic ablation of PPARγ in ILC2s significantly suppresses tumor growth in vivo. Our findings highlight a crucial role for PPARγ in supporting the IL-33 dependent pro-tumorigenic role of ILC2s and suggest that PPARγ can be considered as a druggable pathway in ILC2s to inhibit their effector functions. Hence, PPARγ targeting might be exploited in cancer immunotherapy and in other ILC2-driven mediated disorders, such as asthma and allergy.

[1] Department of Pathology and Immunology, University of Geneva, Geneva, Switzerland. [2] Ludwig Institute for Cancer Research, Lausanne Branch, Lausanne, Switzerland. [3] Department of Oncology UNIL CHUV, University of Lausanne, Lausanne, Switzerland. [4] Department of Pathology and Immunology, Division of Clinical Pathology, University and University Hospitals of Geneva, Geneva, Switzerland. [5] Center for Integrative Genomics, University of Lausanne, Lausanne, Switzerland. [6] Univ. Bourgogne Franche-Comté, INSERM, EFS BFC, UMR1098, Interactions Hôte-Greffon-Tumeur/Ingénierie Cellulaire et Génique, Besançon, France. [7] University Hospital of Besançon, Department of Medical Oncology, Besançon, France. [8] Department of Pharmacy, University of Naples Federico II, Naples, Italy. [9] Institute of Molecular Health Sciences, ETH Zürich, Zürich, Switzerland. [10] Institute of Pathology, University of Bern, Bern, Switzerland. [11] These authors jointly supervised this work: Sara Trabanelli, Camilla Jandus. ✉email: Camilla.jandus@unige.ch

nnate lymphoid cells (ILCs) are the most recently identified subset of innate lymphocytes. ILCs can be classified in three principal groups according to their transcription factor expression and cytokine secretion, mirroring to some extent CD4 T helper subsets[1]. Group 1 ILCs (ILC1s) express T-bet and secrete IFN-γ. Group 2 ILCs (ILC2s) express GATA3 and secrete IL-13 and IL-5. Tissue-resident group 3 ILCs (ILC3s) express RORγT and secrete IL-22 and IL-17 while, in the human peripheral circulation, ILC3s also comprise a population of progenitor cells (ILCPs) able to differentiate into all ILC subsets and natural killer (NK) cells[2]. In the last few years, different reports highlighted the important contribution of ILCs to different physiological and pathophysiologic processes[3]. In particular, ILC2s have been reported to play a critical role in several inflammatory diseases including asthma, chronic rhinosinusitis, and allergic rhinitis[4–6]. Moreover, we and others have shown that ILC2s mediate tumor immune responses, with either pro- or anti-tumor effects depending on the tumor type[7–11]. Nonetheless, external and cell-intrinsic regulatory factors that dictate ILC2 activation and function remain poorly studied, particularly in humans. Better knowledge on ILC2-drivers is warranted to enable pharmacological interference with ILC2 functions in disease settings. In that regard, alarmins (e.g., IL-33) were shown to directly activate ST2-expressing ILC2s and induce their Type-2 cytokine secretion[12]. Further, it was reported that certain types of lipids or lipid-derivatives act as key mediators involved in ILC2 activation. In particular, in the context of helminth infections, ILC2 function is dependent on fatty acid synthesis and oxidation, which is known to be regulated by peroxisome proliferator-activated receptors (PPARs)[13,14]. PPARs are a class of nuclear hormone receptors that act as transcription factors regulating gene expression[15]. They are divided in three major isoforms: α, β/δ, and γ that differ in their tissue expression and functions[15]. PPARγ regulates the transcription of genes associated with lipid metabolism and is expressed in different immune cells, including lymphocytes, monocytes, dendritic cells, and platelets, where it mainly exerts anti-inflammatory effects[16]. Although different synthetic PPARγ ligands are used in the clinics, 15-deoxy-Δ12-14-PGJ2 (15d-PGJ2) is one of the few mediators that is often referred as an endogenous PPARγ ligand[17]. 15d-PGJ2 is a prostaglandin D2 (PGD2)-derived product that has been reported to be essential for ILC2 activation[18–20]. In particular, it has recently been shown that PGD2 and its metabolites (including 15d-PGJ2) are endogenously synthetized after ILC2 stimulation with IL-33, IL-25, and TSLP[19]. Furthermore, elevated levels of PPARγ and other genes involved in prostaglandin (PG) synthesis and response were recently reported in a single cell mRNA-sequencing analysis of ILC2s sorted from human tonsil tissue[21]. In this work, we characterize the expression and functional role of PPARγ in human and mouse ILCs to assess whether PPARγ can be targeted in the context of an ILC-directed immunotherapy.

## Results

### Ex vivo and in vitro expanded human ILC2s specifically express PPARγ.
In order to characterize ILC2s in terms of function and activation, we mined a previously performed RNA-sequencing (RNA-seq) analysis of ILC1s, ILC2s, and ILCPs sorted from healthy donor (HD) PBMCs, and compared their transcriptional signature (ArrayExpress E-MTAB-8494). In particular, focusing on fatty acid metabolism-related genes, that have been associated to ILC2 response during helminthic infection, we found that PPARγ (PPARG) was more expressed in ILC2s, as compared to the other ILC subsets (Fig. 1a, left panel), while no differences in PPARα (PPARA) and PPARβ (PPARD) expression were observed among the subsets (Fig. 1a, right panel), as also confirmed by

qPCR (Fig. 1b). Similar observations were made by the analysis of PPAR expression in in vitro expanded ILC subsets, both at mRNA and protein level (Fig. 1c, d). Given that ILCs are considered as the innate mirrors of CD4 T helper subsets, we analyzed the expression of PPARs in the adaptive counterparts of ILC1s, ILC2s, and ILCPs, i.e., Th1, Th2, and Th17 cells, respectively. PPARγ expression was preferentially observed in human Th2 and Th17 but not Th1 cells upon expansion (Supplementary Fig. 1a, d), in contrast to the mouse where PPARγ protein is expressed predominantly in Th2 cells[22]. Taken together, these findings show specific expression of PPARγ in human ILC2s, which is also maintained during in vitro culture.

### PPARγ sustains Type-2 cytokine secretion in human ILC2s.
To better elucidate the functional role of PPARγ in ILC2s, we analyzed the cytokine production of IL-33/IL-25 (referred as IL) stimulated ILC2s, after pre-incubation with non-toxic concentrations of T0070907, a selective and irreversible PPARγ antagonist[23]. As shown in Fig. 2a, b, ILC2s treated with T0070907 produced significantly less IL-13 as compared to control ILC2s. Moreover, a trend for reduction was also observed for other ILC2-specific cytokines, including IL-4 and amphiregulin (AREG) (Fig. 2b)[24]. These results were confirmed by the quantification of cytokines in cell culture supernatants of ILC2s, pre-treated or not with T0070907 (Fig. 2c). This additional readout allowed us to observe that T0070907 not only reduced the secretion of IL-13 in a concentration-dependent manner (Supplementary Fig. 2a), but also of IL-5 in pre-treated ILC2s (Fig. 2c). To test if the inhibitory effects of T0070907 were due to the selective inhibition of PPARγ, we transfected human in vitro expanded ILC2s with a small interfering RNA (siRNA) targeting PPARγ. The knockdown of PPARγ was confirmed by qPCR analysis (Supplementary Fig. 2b). Consistent with the impact of pharmacological inhibition, PPARγ silencing led to significantly reduced cytokine production and secretion (Supplementary Fig. 2c–e). This effect on Type-2 cytokine secretion might be at least in part a direct consequence of PPARγ binding to peroxisome proliferator response elements (PPREs), since we found motives partially matching the human PPARγ-retinoid X receptor alpha (RXRα) heterodimer binding motif in the promoter region of both IL-5 and IL-13 genes (Fig. 2d, Supplementary Data 1). To substantiate this finding, we performed Chromatin immunoprecipitation (ChIP) on human expanded ILC2s stimulated with the cytokine cocktail and treated or not with T0070907. The results indicated that PPARγ directly bound to the promoter of IL-13 in cytokine-activated ILC2s (Supplementary Fig. 2f). An increasing number of reports demonstrated a direct correlation between PG synthesis and PPARγ activation[25–27]. In particular, it has been recently reported that the inhibition of the cyclooxygenase (COX), which is the key rate-limiting enzyme in PG production, prevents IL-5 and IL-13 secretion by cytokine-stimulated ILC2s[19]. Based on these data, we speculated that COX and PPARγ could exert a synergistic effect on ILC2 cytokine secretion. Hence, we treated ILC2s with a combination of T0070907 and celecoxib (CXB), a selective inhibitor of COX-2, a gene upregulated in ILC2s after cytokine stimulation[19]. Although not statistically significant, we observed that this treatment induced a partial reduction in IL-13 and IL-5 secretion (Supplementary Fig. 2g).

### PPARγ inhibition results in mitochondrial dysfunction in human ILC2s.
As a lipid sensor, PPARγ plays a key role in the regulation of metabolism by affecting, among others, mitochondrial biogenesis, glucose uptake, and lipid intake[28]. To further dissect the mechanisms underlying the PPARγ-mediated

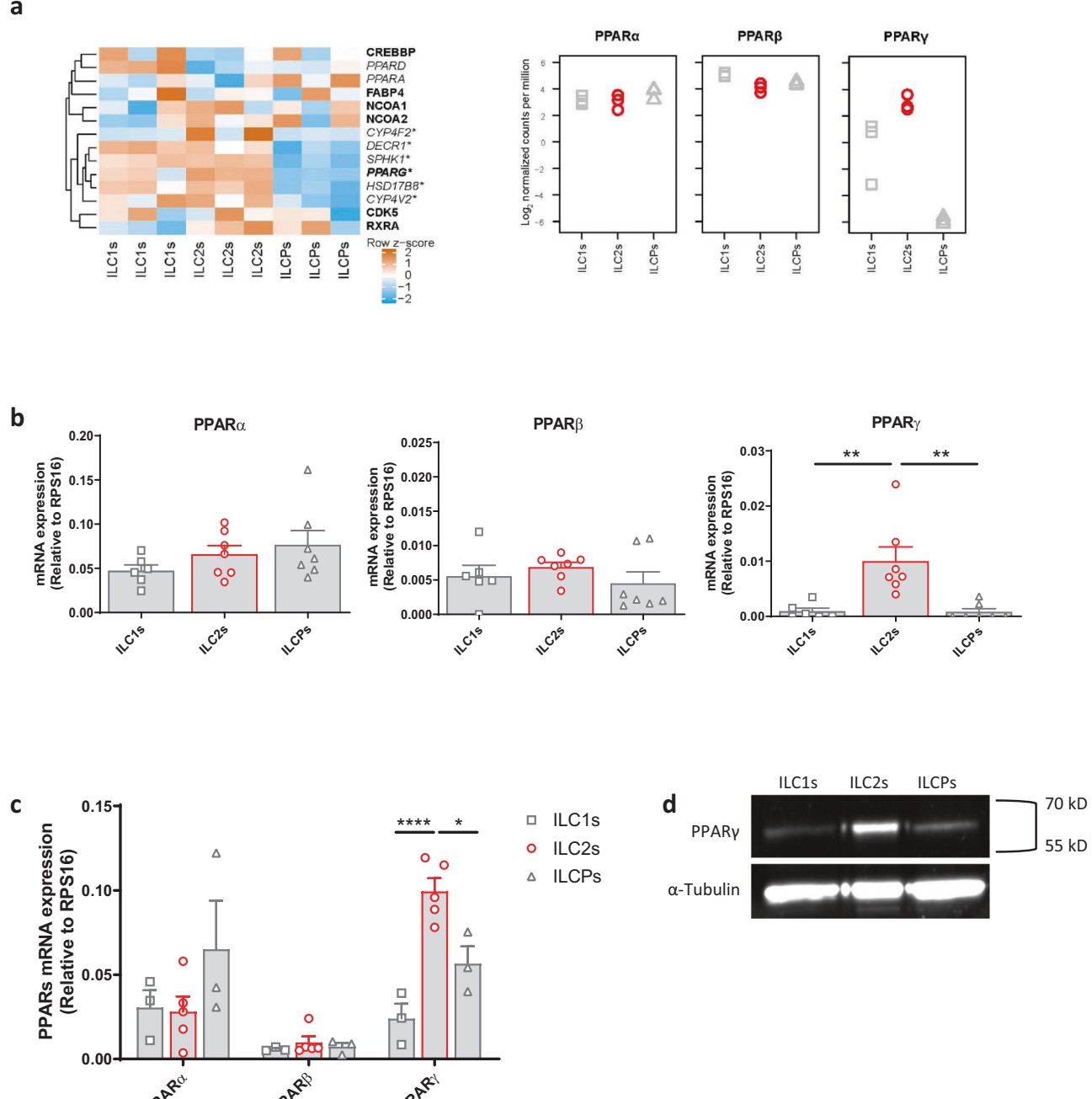

**Fig. 1 Human ex vivo and in vitro expanded ILC2s specifically express PPARγ. a** Heatmap of row z-scores of mRNA level of proteins involved in fatty acid metabolism (highlighted in italics) or involved in protein-protein interaction with PPARγ (highlighted in bold). Fatty acid metabolism genes were taken from the "fatty acid metabolic process" Gene Ontology gene set (GO:0006631, http://geneontology.org/). RNA-sequencing counts were transformed to log2 counts per million, and differential gene expression analysis was performed by fitting a linear model to each gene using the voom function in the limma package for R (v. 3.38.3), followed by a moderated t statistic computation using the empirical Bayes statistics model implemented in the limma package (See Ercolano et al.[1] for detailed methods). Genes significantly upregulated in ILC2s compared to ILC1s and/or ILCPs are indicated using an * ($n = 3$; *$P < 0.05$). Proteins putatively involved in protein–protein interactions were taken from the STRING database (https://string-db.org/). **b** Expression of *PPARs* assessed by qPCR in human freshly sorted ILC subsets (open square ILC1s, open circle ILC2s, open triangle ILCPs) (ILC1s $n = 6$; ILC2s and ILCPs $n = 7$; ILC1s vs ILC2s **$P = 0.0032$, ILC2s vs ILCPs **$P = 0.0021$). **c** Expression of *PPARs* assessed by qPCR in in vitro expanded human ILC subsets (open square ILC1s, open circle ILC2s, open triangle ILCPs) (ILC1s and ILCPs $n = 3$; ILC2s $n = 5$; ILC1s vs ILC2s ****$P < 0.0001$, ILC2s vs ILCPs *$P = 0.0020$). **d** Western blot analysis of PPARγ expression at protein level in ILC2s compared to ILC1s and ILCPs (one individual experiment). Each symbol represents one individual donor. Data are shown as mean ± SEM and were analyzed by one- (**b**) or two-way (**c**) ANOVA tests. Source data are provided as a Source data file.

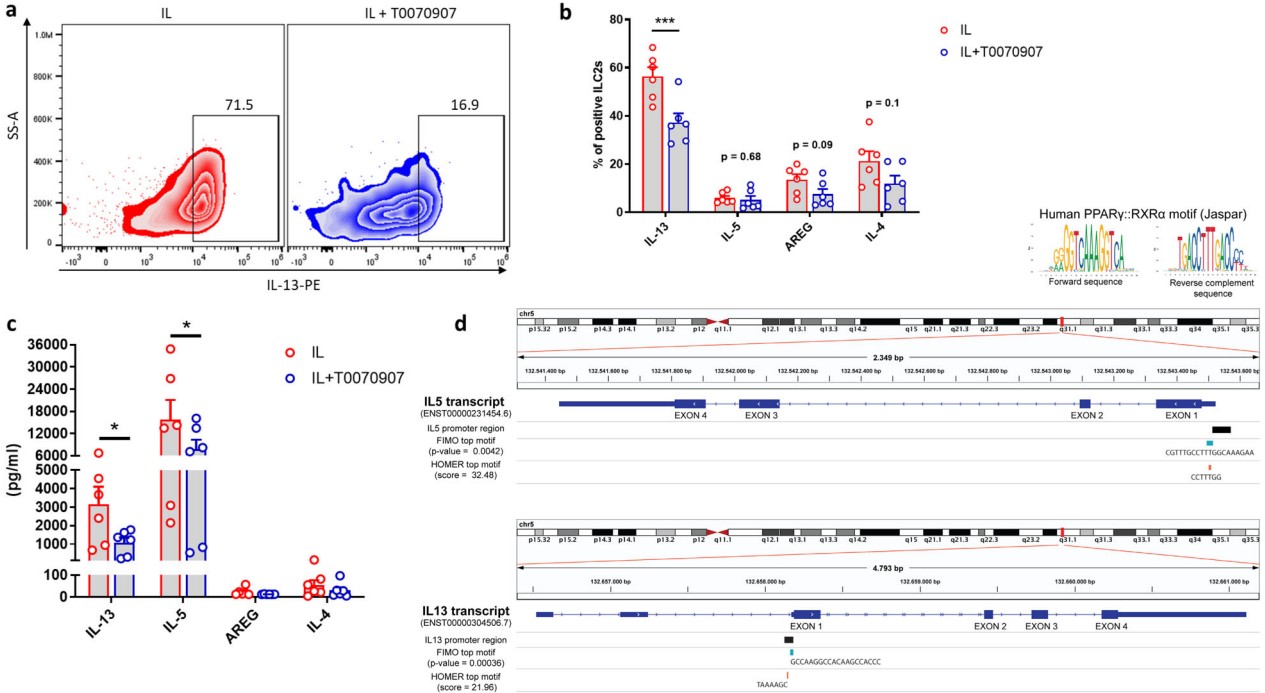

**Fig. 2 PPARγ sustains Type-2 cytokine secretion in human ILC2s. a** Representative example of flow cytometry analysis of ILC2-derived IL-13 upon treatment (blue dot plot) or not (red dot plot) with T0070907. **b** Frequencies of Type-2 cytokine positive cells among ILC2s, after treatment (open blue circle) or not (open red circle) with T0070907 ($n = 6$; ***$P = 0.0002$). **c** Quantification of Type-2 cytokines assessed by Legendplex analysis in cell culture supernatants of ILC2s upon treatment (open blue circle) or not (open red circle) with T0070907 ($n = 6$; IL-13 *$P = 0.048$, IL-5 *$P = 0.040$). **d** Location of motifs similar to the PPARγ-RXRα binding motif (top) in the promoter region of IL-5 and IL-13 (human genome). The motifs with the lowest raw $p$-value found by the FIMO tool (blue) or the highest score found by the HOMER software (orange) are shown. The Integrative Genomics Viewer (IGV, v2.8.0) was used to represent the location of these motifs within the IL-5 and IL-13 promoters[84]. In **b–c** each symbol represents one individual donor. Data are shown as mean ± SEM and were analyzed by two-way ANOVA tests. Source data are provided as a Source data file.

inhibition of ILC2s, we measured mitochondrial mass by Mito-Tracker Green uptake and MitoTracker Deep Red fluorescence in these cells. As shown in Fig. 3a–d, both MitoTracker Green and Deep Red dye uptake were decreased in ILC2s after treatment with the PPARγ inhibitor, suggesting reduced mitochondrial mass. We also used tetramethylrhodamine methyl ester (TMRM) that, by accumulating in intact and active mitochondria, allows to assess changes in their membrane potential. Consistent with the MitoTracker results, the TMRM assay showed a significant decrease of mitochondrial potential after incubation of ILC2s with T0070907 (Fig. 3e, f). To corroborate our data, we performed electron microscopy (EM) analysis that showed reduced percentage of mitochondrial volume per cell in T0070907-treated ILC2s as compared to control (Fig. 3g, h). These findings indicate that the inhibitory effects of T0070907 correlates with a decline in mitochondrial function, which in turn likely impairs ILC2 cellular fitness.

**Tumor-derived IL-33 sensitizes ILC2s to produce IL-13.** Over the last decade, alarmins have been identified as signaling mediators involved in cancer development and progression[29,30]. In particular, different studies reported that IL-33 and its receptor ST2 play a critical role in the pathogenesis of different types of cancers, including breast cancer, hepatocellular carcinoma, and colorectal cancer (CRC)[31–34]. We used CRC as a model to address a putative role of tumor-derived IL-33 on ILC2 functions in the context of tumor immunity. First, we confirmed expression of IL-33 in human CRC tissues using a Tissue Micro Array (Fig. 4a). GATA3$^+$CD3$^-$ cells were identified in proximity to IL-33$^+$ CRC epithelial cells, suggesting that IL-33 released from these malignant cells may stimulate ILC2s in human CRC lesions

(Fig. 4b). In line with these findings, we observed elevated IL-33 levels in the serum of CRC patients as compared to HDs (Fig. 4c). Next, we evaluated ILC2 frequency and function, in terms of cytokine production, in both PBMCs and tumor-infiltrating lymphocytes (TILs) in CRC patients. As previously reported, circulating ILC2 frequency was comparable in HDs and CRC patients (Fig. 4d)[35]. Importantly, ILC2s are present among TILs, as assessed by flow cytometry analysis of dissociated CRC tissues (Fig. 4d). In terms of function, ILC2s from CRC patients' PBMCs produced more IL-13 compared to ILC2s from HDs' PBMCs (Fig. 4e, f). To gain further insight into the role of PPARγ in ILC2s in the CRC setting, we evaluated the expression of *PPARγ*, as well as of *CPT1A*, one of the well-characterized PPARγ-direct target genes[36], in freshly-sorted ILC2s from both PBMCs and TILs of CRC patients. As shown in Supplementary Fig. 3a, b, a trend for higher expression of *PPARγ* and *CPT1A* was observed in CRC patients' PBMCs compared to HDs, suggesting that PPARγ is not only increased but also more active in the CRC setting. In addition, exposure of expanded ILC2s from CRC patients to T0070907 resulted in the reduction of IL-13 and IL-5 secretion as well as in impairment of mitochondrial functions, thus recapitulating the effects observed in HD ILC2s (Supplementary Fig. 3c–e). Collectively, these results indicate that ILC2s from cancer patients are more prone to secrete cytokines, which may be related to their increased exposure to IL-33, both systemically and in the tumor microenvironment.

**IL-13 drives the crosstalk between ILC2s and cancer cells.** In the last few years, IL-13 and its receptors have been identified as novel targets for cancer therapy, and inhibition of IL-13-producing cells as a strategy to reach this goal[37]. To better

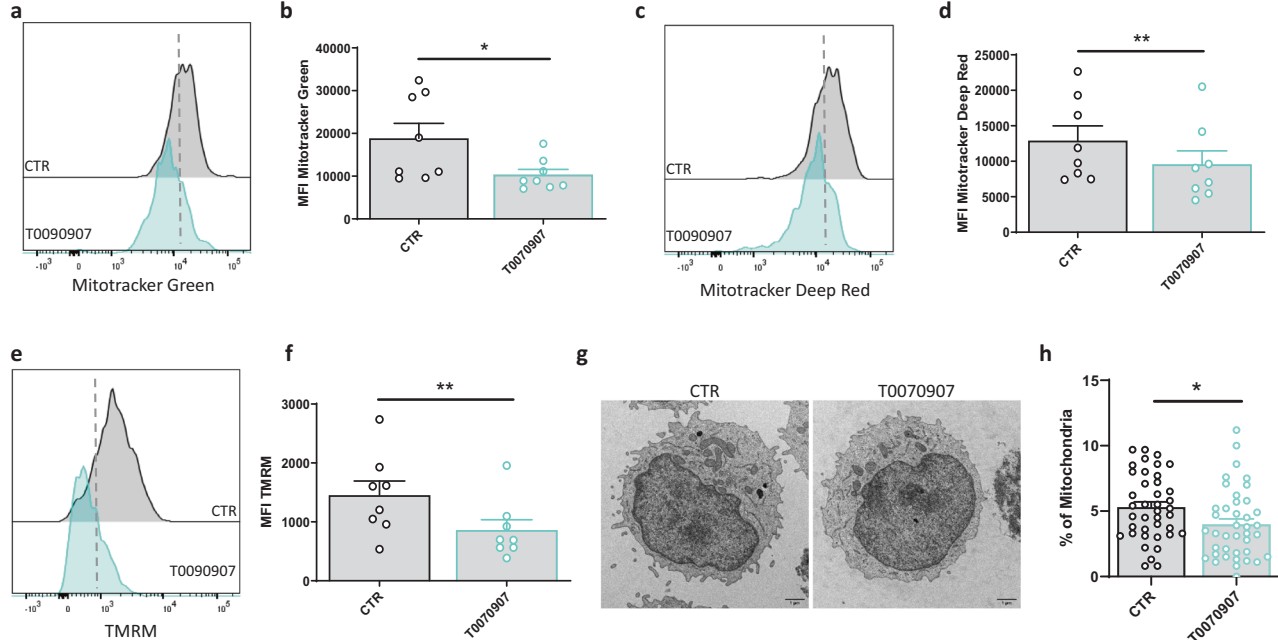

**Fig. 3 PPARγ inhibition in ILC2s results in mitochondrial dysfunction. a, c, e** Representative examples of flow cytometry analysis of MitoTracker Green (**a**) MitoTracker Deep-red (**c**) and TMRM (**e**) in ILC2s untreated (gray histograms) and after T0070907 treatment (blue histograms) for 48 h, with their respective quantification in terms of mean fluorescence intensity (MFI), in **b, d, f** ILC2s untreated (open circle) and after T0070907 treatment (open aqua circle), (n = 8; Mitotracker Green *P = 0.0282, Mitotracker Deep Red **P = 0.0019, TMRM **P = 0.0036). Representative electron microscopy images (×4800 magnification; scale bars: 1 µm) (**g**) and quantitative plots (**h**) of mitochondrial volume in ILC2s untreated (open circle) and after T0070907 treatment (open aqua circle) for 48 h (n = 40 cells examined for each experimental condition over one independent experiment; *P = 0.0307). Each symbol represents one individual donor or one cell. Data are shown as mean ± SEM and were analyzed by two-tailed Wilcoxon tests. Source data are provided as a Source data file.

understand the crosstalk between PPARγ, ILC2s, and the CRC cells, we pre-treated ILC2s with T0070907 or solvent-control and co-cultured them with the SW1116 CRC cell line. As shown in Fig. 5a, b, the ILC2-SW1116 co-culture resulted in increased IL-13 production by ILC2s that was abolished by pre-treating ILC2s with T0070907. IL-13 production in ILC2s was possibly triggered by increased expression of IL-33 in co-cultured CRC cells (Fig. 5c). To address this hypothesis, we performed co-culture experiments using the helminth parasite-secreted protein (HpARI), known to suppress type 2 immune responses through interference with the IL-33 pathway[38]. As shown in Supplementary Fig. 4a, the addition of HpARI in the medium significantly reduced the production of IL-13 by ILC2s, after co-culture with SW1116 colorectal cancer cells. Since IL-13 is involved in CRC progression and metastasis development by affecting epithelial to mesenchymal transition (EMT)[39], we hypothesized that ILC2-derived IL-13 could affect EMT in CRC cells. To address this point, we performed wound healing and clonogenic assays using conditioned medium (CM) of activated ILC2s, pre-treated or not with T0070907 (ILC2 CM and ILC2 + T0070907, respectively). No effect was observed on the CRC cell proliferation (Supplementary Fig. 4b). However, we found that the presence of ILC2 CM increased the migration of SW1116 cells, whereas addition of ILC2 + T0070907 CM lead to the opposite effect (Fig. 5d, e). Likewise, the colony formation assay confirmed that ILC2 CM increased the number of SW1116 colonies compared to control and to ILC2 + T0070907 CM (Fig. 5f). Moreover, the addition of an anti-IL-13 blocking antibody abrogated the development of SW1116 colonies, confirming the involvement of IL-13 in the ILC2-PPARγ dependent effect on CRC cells (Fig. 5f). In order to corroborate our findings on the pro-tumor effects of ILC2 CM for the migration and invasion of

CRC, we evaluated in the SW116 line the expression of *MMP9* and *N-cadherin*, two molecules that have been linked to the progression and invasion of tumor cells[40–42]. We found that the addition of ILC2 CM significantly increased the expression of these two EMT markers on CRC cells, whilst the addition of ILC2 + T0070907 CM or the anti-IL-13 blocking antibody reduced their expression towards basal levels (Fig. 5g). Taken together our data suggest a crosstalk between cancer cells and ILC2s, in which cancer cells trigger ILC2s to produce IL-13, which in turn sustains their migratory and invasive capacity, yet without affecting their proliferation rate. Inhibition of PPARγ in this circuit thwarts the pro-tumorigenic role of ILC2s.

**PPARγ is expressed and functional in murine ILC2s.** To translate our findings into an in vivo setting, we assessed the expression of *PPARg* in freshly-sorted ILC subsets from the lung of C57BL/6 mice. In agreement with our human data, *PPARg* was specifically expressed in ILC2s, while it was not detectable in ILC1s and ILC3s (Fig. 6a). Therefore, we exposed murine ILC2s to T0070907 in vitro and could thereby recapitulate the results obtained using human ILC2s, with a reduction in the production of both IL-13 and IL-5 (Fig. 6b, c). Next, to define the role of PPARγ in ILC2s beyond its pharmacological inhibition, we used a transgenic mouse model with an ILC-specific PPARγ inducible knockout (KO) by crossing *Pparg*fl/fl animals with mice expressing the Cre enzyme under the Id2 promoter, known to control ILC development[43]. This model allows tamoxifen-mediated induction of Cre activity in time and in an ILC-specific manner[44] (Fig. 6d). We next sensitized *Pparg*fl/fl*Id2*CreER^T2 positive or negative mice (also reported as *Pparg*fl/fl *Id2*Cre+ and *Pparg*fl/fl *Id2*Cre, respectively) with IL-33 and IL-25 i.p. for 3 days, after 5 days of tamoxifen injection. As shown in Fig. 6e, ILC2s

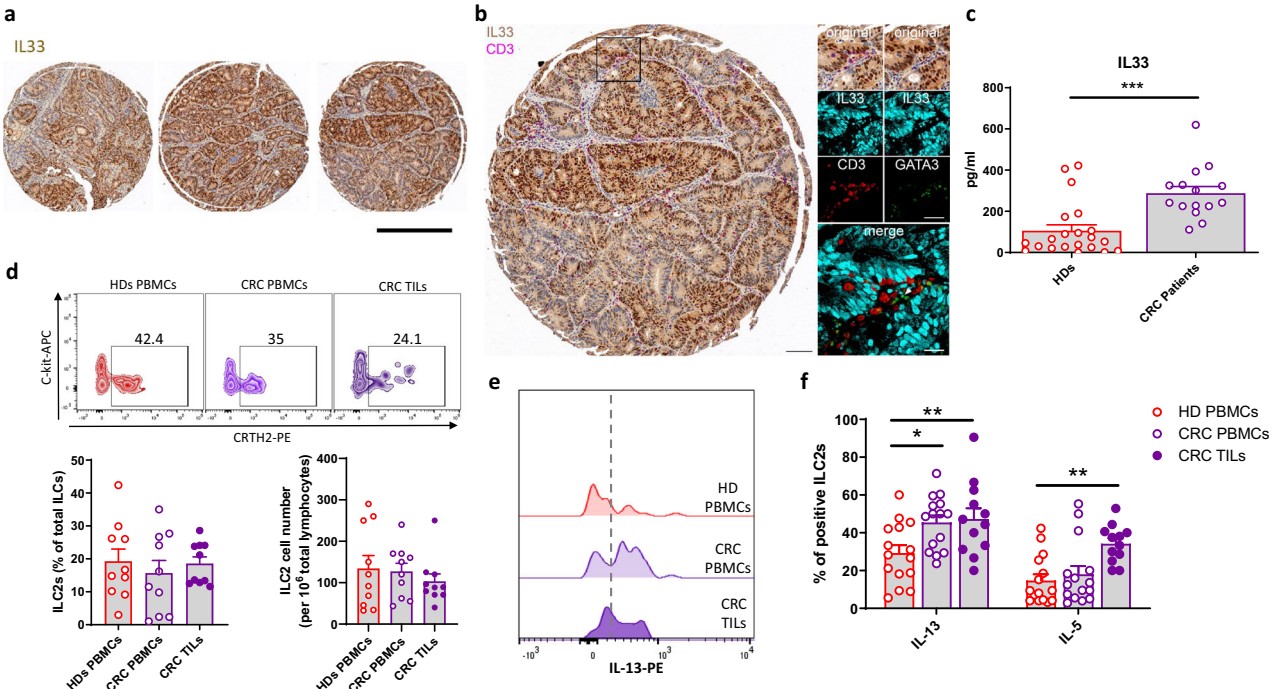

**Fig. 4 Tumor-derived IL-33 sensitizes ILC2s to produce IL-13. a** Representative examples of IL-33+ tumor epithelium in human CRC tissues (one out of 88 stained cores in a CRC tissue microarray) Scale bar: 600 μm. **b** Representative examples of CRC core stained for IL33 and CD3 (big insert), a magnified view of the area marked by the black square after both staining have undergone color deconvolution and co-registration (right insert). Scale bars: Big insert: 100 μm small tiles: 50 μm merge (big tile): 25 μm. **c** Quantification of IL-33 plasma levels assessed by ELISA in CRC patients (open violet circle) and healthy donors (open red circle) (HDs) (HD $n = 21$, CRC Patients $n = 15$; ***$P = 0.0001$). **d** Representative examples and quantification in terms of frequency and absolute numbers of ILC2s in HD PBMCs (open red circle), CRC patients' PBMCs (open violet circle) and TILs (closed violet circle) ($n = 10$). **e** Representative example of flow cytometry analysis of ILC2-derived IL-13 in HD PBMCs, CRC patients' PBMCs or TILs. **f** Frequencies of IL-13 and IL-5 positive ILC2s in ex-vivo HD PBMCs (open red circle), CRC patients' PBMCs (open violet circle) or TILs (closed violet circle) (HDs and CRC PBMCs $n = 15$, CRC TILs $n = 12$; IL-13 HD PBMCs vs CRC PBMCs *$P = 0.0144$, IL-13 HD PBMCs vs CRC TILs **$P = 0.0094$, IL-5 HD PBMCs vs CRC TILs **$P = 0.0044$). Each symbol represents one individual donor. Data are shown as mean ± SEM (*$P < 0.05$; **$P < 0.01$; ***$P < 0.001$) and were analyzed by Wilcoxon (**c**), one (**d**) - or two-way (**f**) ANOVA tests. Source data are provided as a Source data file.

sorted from *Pparg*fl/fl *Id2*Cre+ mice secreted significantly less IL-13 and IL-5 compared to control *Pparg*fl/fl *Id2*Cre− mice. In addition, similarly to the results observed in humans, we found motives partially matching the mouse PPARγ-Rxrα heterodimer binding motif in the promoter region of both *Il-5* and *Il-13* genes (Fig. 6f, Supplementary Data 2). Taken together these results confirm the role of PPARγ as a key regulator of ILC2 function in vivo.

**PPARγ expression in ILC2s influences cancer progression in a CRC murine model.** Next, to investigate the in vivo role of ILC2s in CRC development and progression, we used a heterotopic murine model of CRC in *RORa*fl/sg*Il7r*Cre mice (referred to as ILC2 KO mice)[45]. We injected subcutaneously (s.c.) MC-38 murine genetically modified CRC cells to constitutively produce IL-33 (reported as MC-38-IL33). Tumor cells were injected into the right flank of ILC2 KO and littermate control mice, and survival was monitored. As displayed in Fig. 7a, tumor progression was slower in ILC2 KO mice compared to littermate controls, resulting in better survival advantage in the KO strain, suggesting a pro-tumorigenic function of ILC2s in this model. This effect was reverted after the adoptive transfer of expanded ILC2s sorted from the lungs of C57BL6 mice (Fig. 7b). Next, to further assess the contribution of PPARγ to ILC2s during tumorigenesis, we injected MC-38-IL33 cells into *Pparg*fl/fl*Id2*Cre+ mice and *Pparg*fl/fl *Id2*Cre− mice (Fig. 7c).

Strikingly, after 18 days, tumor volume was reduced in *Pparg*fl/fl *Id2*Cre+ mice compared to control (Fig. 7d). In line with these data, *Pparg*fl/fl*Id2*Cre+ mice showed a longer survival compared to control mice (Fig. 7e). To confirm our in vitro results on the involvement of ILC2s in the EMT phenomenon, we evaluated by qPCR analysis the expression of murine MMP-9 and N-cadherin in dissociated tumor tissues. As shown in Fig. 7f, higher expression of both *Mmp9* and *Ncad* in *Pparg*fl/fl*Id2*Cre− compared to *Pparg*fl/fl*Id2*Cre+ tumors were observed. In addition, to evaluate the therapeutic potential of PPARγ inhibition in CRC development and progression, we treated tumor-bearing mice with T0070907. As shown in Supplementary Fig. 6a, treatment with T0070907 significantly reduced both tumor volume and weight, suggesting a possible therapeutic use of PPARγ inhibitors in CRC patients. Furthermore, we also assessed the pro-tumor effects of IL-13 in vivo by treating tumor-bearing mice with an anti-IL-13 neutralizing antibody. After 18 days of treatment, both tumor volume and weight were significantly reduced in treated mice compared to controls (Supplementary Fig. 6b). Taken together, these results underscore the relevance of ILC2s and PPARγ in fostering cancer cell progression and subsequent metastatic potential.

**Discussion**

ILC2s are emerging as key regulators of Type-2 immune responses, by influencing downstream adaptive immunity in

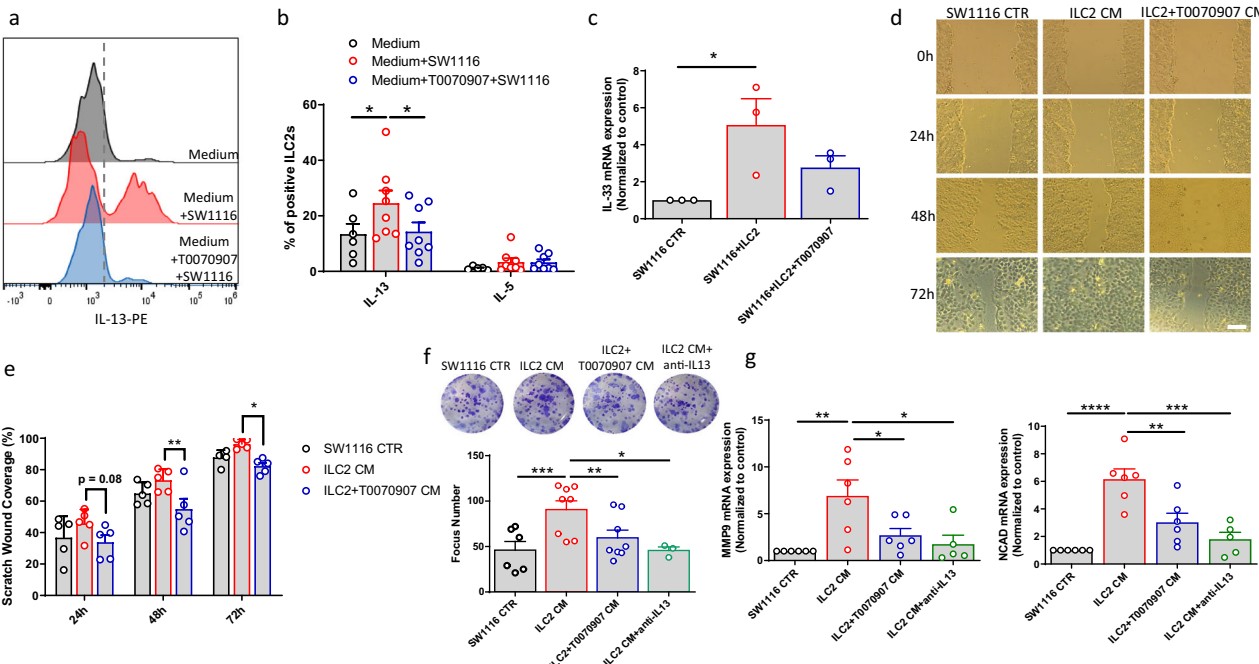

**Fig. 5 IL-13 drives the crosstalk between ILC2s and cancer cells. a** Representative example of flow cytometry analysis of ILC2-derived IL-13 upon co-culture with SW1116 CRC cells. **b** Frequencies of IL-13 and IL-5 positive ILC2s with T0070907 upon co-culture with SW1116 CRC cells ((Medium (open circle) Medium + SW1116 (open red circle) Medium + T0070907 + SW1116 (open blue circle)) (Medium $n = 6$, Medium+SW1116 $n = 8$, Medium + T0070907 + SW1116 $n = 8$; Medium vs Medium + SW1116 *$P = 0.0313$, Medium + SW1116 vs Medium + T0070907 + SW1116 *$P = 0.0103$). **c** Expression of IL-33 assessed by qPCR analysis in SW1116 CRC cells upon co-culture with ILC2s (SW1116 CTR (open circle) SW1116 + ILC2 (open red circle) SW1116 + ILC2 + T0070907 (open blue circle) ($n = 3$; *$P = 0.0419$). **d** Representative example of a wound healing assay (×20 magnification; Scale bar: 50 μm) performed using SW1116 CRC cells (CTR (open circle)) after incubation with ILC2 (open red circle) or ILC2 + T0070907 conditioned medium (CM) (open blue circle). **e** Quantification of the healed wound area at 24, 48, and 72 h ($n = 5$; **$P = 0.0078$, *$P = 0.0493$). **f** Representative example and quantification of clonogenic assays performed using SW1116 CRC cells (open circle) after incubation with ILC2 CM (open red circle), ILC2 + T0070907 CM (open blue circle) or ILC2 CM + anti-IL-13 blocking antibody (open green circle) for 14 days (SW1116 CTR $n = 6$, ILC2 CM and ILC2 + T0070907 CM $n = 8$, ILC2 CM + anti-IL13 $n = 3$; ***$P = 0.0002$, **$P = 0.0085$, *$P = 0.0497$). **g** Expression of *MMP9* (left) and *N-cadherin* (right) assessed by qPCR analysis in SW1116 CRC cells (open circle) upon incubation with ILC2 CM (open red circle), ILC2 + T0070907 CM (open blue circle) or ILC2 CM + anti-IL-13 blocking antibody (open green circle) (SW1116 CTR, ILC2 CM, ILC2 + T0070907 CM $n = 6$, ILC2 CM + anti-IL13 $n = 5$; *MMP9* SW1116 CTR vs ILC2 CM **$P = 0.0035$, ILC2 CM vs ILC2 + T0070907 CM *$P = 0.0425$, ILC2 CM vs ILC2 CM + anti-IL13 *$P = 0.0152$; *NCAD* SW1116 CTR vs ILC2 CM ****$P < 0.0001$, ILC2 CM vs ILC2 + T0070907 CM **$P = 0.0041$, ILC2 CM vs ILC2 CM + anti-IL13 ***$P = 0.0002$). Each symbol represents one individual donor or sample. Data are shown as mean ± SEM and were analyzed by one- (**c**, **f**, **g**) or two-way (**b**, **e**) ANOVA tests. Source data are provided as a Source data file.

homeostasis and disease. However, both extrinsic and cell-intrinsic regulatory pathways dictating ILC2 functions remain elusive. Here, we demonstrate that PPARγ is selectively expressed in ILC2s, both in humans and mice. Its pharmacological inhibition impairs mitochondrial fitness and reduces Type-2 cytokine secretion in ILC2s upon IL-33 stimulation. Furthermore, in the environment of a human IL-33-enriched cancer type, such as CRC, ILC2s show PPARγ-dependent pro-tumoral functions. Lastly, by inducible and conditional deletion of PPARγ in ILCs using *PPARg*[fl/fl]*Id2*CreER[T2] mice and a heterotypic tumor model, we show significant reduction of cancer development and progression in vivo. Overall, our results point to a PPARγ-driven pro-tumorigenic role of ILC2s in IL-33-dependent tumors and suggest that PPARγ targeting in ILCs may become an attractive add-on therapy for ILC2-driven diseases, including cancer.

It has been reported that fatty acid metabolism-related genes (including hexokinase 2 (HK2), pyruvate dehydrogenase kinase 1 (PDK1), and FATP-6 play a key role in ILC2 functions during helminthic infections[13]. In other immune cell types, PPARγ was shown to act as a master regulator of lipid metabolism in alveolar macrophages[46], as inhibitor of pro-inflammatory cytokine secretion[47], and to promote type-2 responses[48]. In that regard,

Nobs et al. found that IL-33 drives Th2 responses and the development of pulmonary allergic inflammation by induction of PPARγ in CD4[+] T cells and DCs[22]. More recently, it has been reported that Th9 function is positively regulated by PPARγ, which promotes IL-9 secretion in acute allergic skin diseases[49]. Different ILC transcriptomic analysis previously reported on PPARγ mRNA expression in ILC2s, in both human and mouse tissues[21,50]. Moreover, it has been recently reported that PPARγ is an important regulator of ILC2 during allergic airway inflammation[51,52]. In addition, Batyrova et al., demonstrated that the expression of PD1 on ILC2s is controlled by PPARγ and affects the production of IL-5 and IL-13 in vivo[53]. However, the functional relevance of this nuclear receptor in ILC2s is still limited, particularly in humans. In contrast to the activation-induced PPARγ expression in CD4[+] Th2 cells[22], our results argue for a steady-state expression of PPARγ in ILC2s, that apparently acts as an homeostatic regulator of Type-2 cytokine secretion, likely in the presence of endogenous ligands such as PGD2-derived 15d-PGJ2, as it was previously reported[19]. It has been demonstrated that PPARγ stimulation, via the induction of PGC-1α, promotes mitochondrial biogenesis in different cell types[54]. Our findings are consistent with a role of PPARγ in the

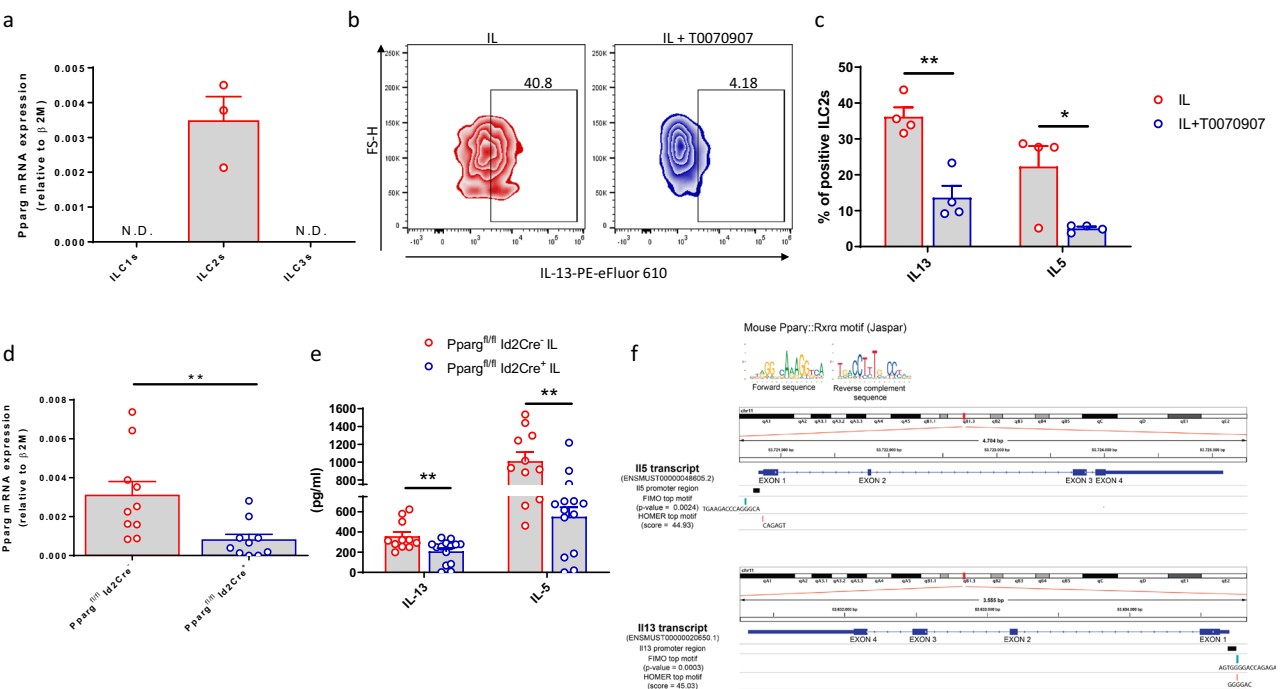

**Fig. 6 PPARγ is expressed and functional in mouse ILC2s. a** Expression of *Pparg* assessed by qPCR analysis in freshly-sorted ILC subsets from the lung of C57BL6 mice ($n = 3$). **b** Representative example of flow cytometry analysis of mouse ILC2-derived IL-13 upon T0070907 treatment in vitro. **c** Frequencies of IL-13 and IL-5 positive ILC2s untreated (open red circle) and after T0070907 treatment (open blue circle) ($n = 4$; *$P = 0.0101$, **$P = 0.0015$). **d** Expression of *Pparg* assessed by qPCR analysis in freshly-sorted ILC2s from the lung of IL-33/IL-25-treated *Pparg*fl/fl*Id2*CreER^T2 positive (open blue circle) or CreER^T2 negative mice (open red circle) ($n = 10$, **$P = 0.0098$). **e** Quantification of IL-13 and IL-5 assessed by Legendplex analysis in cell culture supernatants of murine ILC2s upon stimulation with IL-33 and IL-25 in vitro (*Pparg*fl/fl*Id2*CreER^T2 positive (open blue circle) or CreER^T2 negative mice (open red circle)) (*Pparg*fl/fl*Id2*CreER^T2 negative $n = 11$, *Pparg*fl/fl*Id2*CreER^T2 positive $n = 14$; IL-13 **$P = 0.0086$, IL-5 **$P = 0.0035$). **f** Location of motives similar to the PPARγ-RXRα binding motif (top) in the promoter region of IL-5 and IL-13 (mouse genome). The motives with the lowest raw p-value found by the FIMO tool (blue) or the highest score found by the HOMER software (orange) are shown. The Integrative Genomics Viewer (IGV, v2.8.0) was used to represent the location of these motifs within the IL-5 and IL-13 promoters[84]. See Supplementary Data 2 for complete motif search results. Each symbol represents one individual mouse. Data are shown as mean ± SEM and were analyzed by Wilcoxon (**d**) or two-way (**c**, **e**) ANOVA tests. Source data are provided as a Source data file.

maintenance of energetic homeostasis also in ILC2s, possibly by boosting PGC-1α expression and promoting mitochondrial activity. Further, the expression of IL-13 and IL-5 could be directly regulated by the PPARγ-RXRα heterodimer in Type-2 cytokine epigenetically-poised ILC2s[55], as suggested by the presence of potential PPARγRXRα binding elements in the promoter regions of these genes. Whether or not PPARγ expression is heterogeneous in distinct developmental stages or/and in subpopulations of ILC2s across tissues remains to be elucidated by high-definition screening methods, such as single-cell mRNA sequencing.

Accumulating evidence has indicated that IL-33 and its receptor ST2 (also known as IL1RL1) represent a key inflammatory pathway in tumor biology and immunology[34,56]. In particular, several studies reported that IL-33 and ST2 are implicated as potent modulators of the TME promoting immune cells recruitment and tumor malignancy in different types of cancer, including breast cancer, hepatocellular carcinoma, gastric cancer, and CRC[31,57,58]. In CRC, different reports showed that IL-33 and ST2 are highly expressed in tumor tissues and correlated with tumor development[31] or advanced tumor stage and poor prognosis[59–61]. ILC2 tumor infiltration has been reported by us and others in different cancer types, being linked with both pro- and anti-tumor activities depending on the context[10]. Signaling through ST2 can directly stimulate Type-2 cytokine production by ILC2s[62]. In turn, ILC2-derived, IL-33-triggered IL-13 was shown to favor myeloid-derived suppressor cells (MDSCs)

induction and to be linked to recurrence in bladder cancer[7,57,63]. Moreover, IL-13 has been reported to promote EMT via IL-13R/STAT6 signaling, but nothing was previously reported on a possible link between EMT and ILCs[39]. Our results show that ILC2s exert an essential role in the tumor-driven IL-33/ST2/IL-13 axis, by promoting the migration and invasion of CRC cancer cells, which are key factors promoting metastasis in vivo[64]. Our results indicate that PPARγ is key in this interaction between ILC2s and cancer cells, by supporting ILC2s in their pro-tumoral functions, that might be conserved across different IL-33 dependent tumors, such as breast and liver cancer. However, in other tumor types, ILC2s might exert anti-tumor functions, thus arguing that the benefit of targeting ILC2s is tumor type-dependent[65]. To date, conflicting data have been reported on PPARγ in cancer development and progression[66]. On the one hand, PPARγ expression has been associated with good prognosis in CRC[67] and the use of the PPARγ agonists troglitazone and rosiglitazone inhibited tumorigenesis in CRC and bladder cancer models by a tumor-directed effect[68,69]. On the other hand, in *Apc*^min mice the same treatment increases the number of tumors, and the use of PPARγ inhibitors (including T0070907) suppressed breast cancer cell proliferation in vitro[70] and restrained CRC cell migration and invasion both in vitro and in vivo[71,72]. In our setting of PPARγ targeting in ILC2s, the observed antitumoral phenotype obtained by the in vivo conditional and inducible genetic deletion of PPARγ in ILCs suggests that ILC-specific and temporal targeting of PPARγ might beneficial in

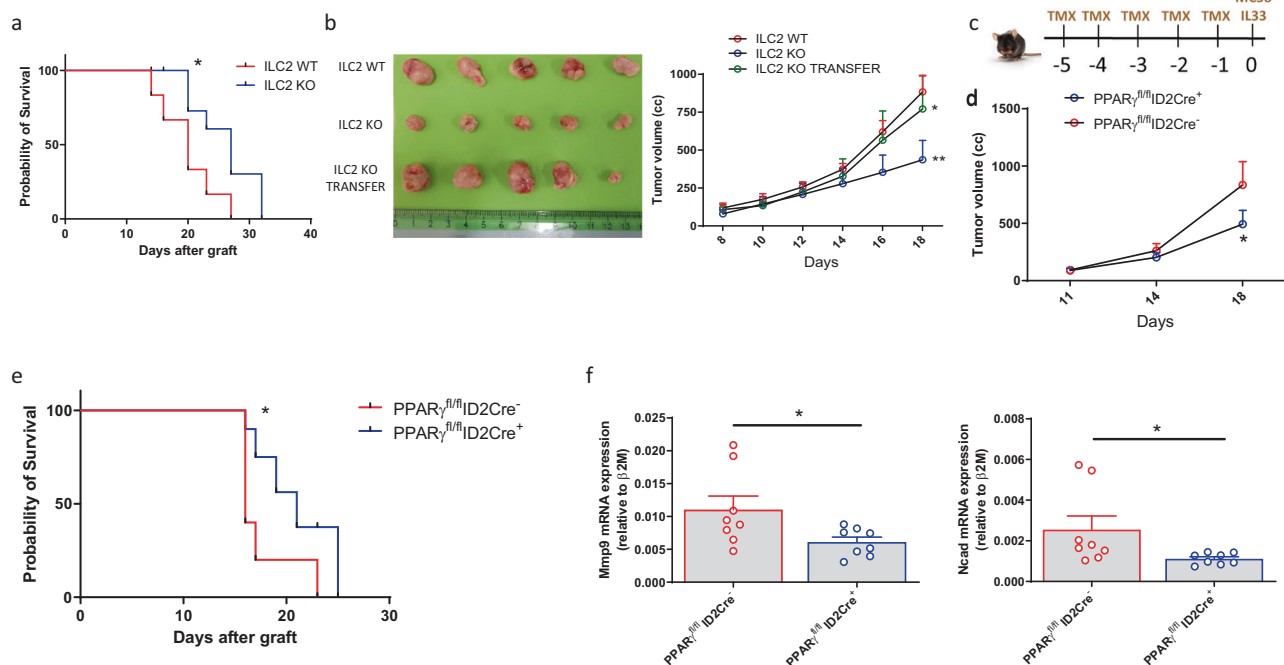

**Fig. 7 PPARγ expression in ILC2s influences cancer progression in a CRC murine model. a** Tumor survival of $RORa^{fl/sg}Il7rCre$ positive (blue bar) or negative (red bar) mice after implantation of MC-38-IL33 murine CRC cells (ILC2 WT $n = 6$, ILC2 KO $n = 7$; *$P = 0.0227$). **b** Representative image of tumor size (left) and tumor development (right) expressed as tumor volume in ILC2 WT (open red circle), ILC2 KO (open blue circle) and ILC2 KO mice ILC2-transferred (open green circle) ($n = 6$; *$P = 0.0473$, **$P = 0.0023$). **c** Schematic representation of CRC murine model in $Pparg^{fl/fl}Id2CreER^{T2}$ mice. **d** Tumor development expressed as tumor volume in $Pparg^{fl/fl}Id2CreER^{T2}$ positive (open blue circle) or negative mice (open red circle) ($n = 15$; *$P = 0.0451$). **e** Survival of $Pparg^{fl/fl}Id2CreER^{T2}$ positive (blue bar) or negative (red bar) tumor-bearing mice ($n = 5$; *$P = 0.0338$). **f** Expression of $Mmp9$ (left) and $Ncad$ (right) assessed by qPCR analysis in $Pparg^{fl/fl}Id2CreER^{T2}$ positive (open blue circle) or negative (open red circle) dissociated tumor tissues ($n = 8$; $Mmp9$ *$P = 0.0379$, $Ncad$ *$P = 0.0104$). Each symbol represents one individual mouse or sample. Data are shown as mean ± SEM and were analyzed by Log-rank (Mantel–Cox) (**a**, **e**) Wilcoxon (**f**) or two-way (**b**, **d**) ANOVA tests. Source data are provided as a Source data file.

cancer. Thus, we identified PPARγ as a druggable target to suppress the potent and rapid ILC2 effector functions. Its pharmacologic inhibition might be used as an add-on therapy for cancer treatment and/or in other ILC2-mediated diseases, as recently demonstrated in allergy and asthma[22,48,73].

## Methods

**Human peripheral blood cell collection.** Venous blood was drawn from healthy donors (HDs) at the Swiss Transfusion Center CRS of Lausanne, Volunteers are asked to read an information sheet for blood donation and to complete an online medical questionnaire on the day of donation. Once the questionnaire finalized, a PDF file is generated for printing, and it is signed to give the approval of the blood donation. PBMCs were isolated by Lymphoprep (Stemcell) centrifugation and immediately cryopreserved.

**CRC patients.** Donors of CRC tissues gave written informed agreement and analyses of the samples were approved by the Cantonal Ethics Committee of Bern (2017-01821 and 2018-01502). Leftover material from diagnostic was used for the tissue microarray (TMA), no extra tissue was collected specifically for this study. Metastatic colorectal cancer patients were included in the "Epitopes-CRC01" (NCT02838381) and "Epitopes-CRC02" (NCT02817178) trials, two French multicentric prospective studies. Epitopes CRC02 ($n = 256$, NCT02817178) and Epitopes CRC01 (ongoing, NCT02838381) are observational studies performed on first-line metastatic colorectal cancer patients. Given that peripheral blood and tissue sampling was performed in both studies, they were retrospectively registered as interventional at ClinicalTrials.gov. However, the patients didn't receive any additional intervention than standard of care. These two studies involved the collection of PBMC and tumor tissue. Epitopes CRC01 included several observational cohorts while Epitopes CRC02 was constituted by only one observational cohort. Samples included in this study were either from cohort A (patient with metastatic colorectal cancer with first-line therapy by chemotherapy +/− surgery and with a disease-free survival > or = at 20 months) of the Epitopes CRC01 study, or from the Epitopes CRC02 study. All patients were enrolled after the signature of an informed consent, in accordance with the French regulation and after the approval by the local and national ethics committee (CPP Grand Est II). Data were

anonymized. For each patient from "Epitopes-CRC02" (NCT02817178) trial, blood samples were collected before any metastatic cancer specific treatment. For patients from "Epitopes-CRC01" (NCT02838381), tumor samples were excised by surgeons from the department of digestive surgery at University Hospital Besançon.

The generation of TIL cultures was based on the methodology previously established by Dudley et al.[74].

We have complied with all relevant ethical regulations for work with human participants, and we confirm that informed consent was obtained from each donor.

**Flow cytometry analysis.** Human ILCs were identified as lineage (Lin) negative and CD127 positive cells. Lineage markers, all FITC-conjugated, include: anti-human CD3 (UCHT1, Beckman Coulter (BC) 1:50), anti-human CD4 (SFCI12T4D11, BC, 2:50), anti-human CD8 (MEM- 31, Immunotools, 1:50), anti-human CD14 (RMO52, BC, 2:50), anti-human CD15 (80H5, BC, 2:50), anti-human CD16 (3G8, BC, 1:100), anti-human CD19 (J3-119, BC, 1:100), anti-human CD20 (2H7, Biolegend, 1:100), anti-human CD33 (HIM3-4, Biolegend, 1:50), anti-human CD34 (561, Biolegend, 1:100), anti-human CD203c (E-NPP3, 1:50) (NP4D6, Biolegend, 2:50), anti-human FcεRIα (AER-37, Biolegend 2:50) anti-human CD56 (REA196, Miltenyi, 1:50). Additional markers used include: Brilliant Violet 421 anti-human CD127 (IL-7Rα) (A019D5, Biolegend, 1:100), APC anti-human CD117 (cKit) (YB5.B8, BD Bioscience, 1:50), PE anti-human CRTH2 (CD294) (BM16, Biolegend, 1:50). Dead cells were excluded using the viability dye Live/Dead Aqua or Vivid IR (Invitrogen). Representative gating strategy is included in Supplementary Fig. 7a. Intracellular staining was performed after fixation and permeabilization with 0.1% saponin (Sigma), using PE-Cy7 (JES10-5A2, Biolegend, 1:50) or PE anti-human IL-13 (7118582, BD Bioscience, 2:50) Alexa Fluor 700 anti-human AREG (In-house, 2:50) PE anti-human IL-4 (11B11, Biolegend, 1:50) and Brilliant Violet 421 anti-human IL-5 (TRFK5, Biolegend, 1:50). Samples were acquired on Gallios flow cytometer (Beckman Coulter) or LSR-II (Becton-Dickinson, San Jose, CA) and data were analyzed using FlowJo software (TreeStar V.10). For ILC isolation, aliquots of cells were sorted to 98% purity using a FACSAria (Becton Dickinson). For mitochondrial activity, ILC2s from healthy donors were stained by: MitoTracker Green (M7514 ThermoFisher), MitoTracker Deep Red (M22426, ThermoFisher), TMRM (T668, ThermoFisher).

Human CD4$^+$ T cell subsets were isolated by FACS using the following antibodies: Brilliant Violet 605 anti-human CD3 (OKT3, Biolegend, 1:100), Alexa Fluor 700

anti-human CD4 (RPA-T4, Biolegend, 1:50), Brilliant Violet 785 anti-human CD45RO (UCHL1, Biolegend, 1:50), PE-Cy7 anti-human CXCR3 (1C6, Biolegend, 2:50), PE anti-human CRTH2 (BM16, Biolegend, 1:50), APC anti-human CD196 (CCR6) (G034E3, Biolegend, 2:50). Cells were gated on alive CD3+CD4+CD45RO+ cells and the Th subsets were sorted as follow: Th1 as CXCR3+CRTH2−CCR6−, Th2 as CRTH2+, and Th17s as CXCR3−CRTH2−CCR6+.

Murine ILCs were identified as CD45+Lin−CD90.2+ among alive cells. Lineage markers, all FITC conjugated, include: anti-mouse CD3e (17A2, in house, 1:200), CD5 (53.7, in house, 1:200), CD19 (ID3, in house, 1:200), CD11b (M1/70, in house, 1:200), CD11c (N418, in house, 1:200), B220 (RA3-6B2, in house, 1:200), CD49b (DX5, Miltenyi Biotec, 1:200), FCεRIa (MAP-1, Miltenyi Biotec, 1:200), Ter119 (Ter119, in house, 1:200), TCRγδ (2M31/11, in house, 1:200) and TCRαβ (H57, in house, 1:200). Additional markers used to identify the ILC subpopulations include: Alexa-Fluor 700 anti-mouse CD45.2 (AL1-4A2, in house, 1:400), PE anti-mouse ST2 (RMST2-2, Invitrogen, 1:200), Brilliant Violet 605 anti-mouse CD90.2 (53-2.1, Biolegend, 1:50), PE-Cy7 anti-mouse KLRG1 (2F1/KLRG1, Biolegend, 1:200), APC anti-mouse CD117 (cKit) (2B8, eBioscience, 1:200), BV510 anti-mouse NKp46 (29A1.4 Biolegend, 1:100), BV711 anti-mouse NK1.1 (PK136, Biolegend, 1:100), BUV395 anti-mouse CD4 (GK1.5, BD, 1:200). Representative gating strategy is included in Supplementary Fig. 7b. Intracellular staining was performed after fixation and permeabilization with 0.1% saponin (Sigma), using Brilliant Violet 421 anti-mouse IL-5 (TRFK5, Biolegend, 1:50) and PEefluor610 anti-mouse IL-13 (4311635, eBioscience, 1:100).

**Cell culture**. The murine colon cancer cell line MC-38 was cultured in complete DMEM GlutaMAX (Gibco) supplemented with 10% FCS (Gibco), 5% penicillin/ streptomycin (Thermo-Fisher), 5% HEPES (Thermo-Fisher), 0.05 mM 2β-mercaptoethanol (Thermo-Fisher). Human SW1116 colon cancer cells were cultured in RPMI-1640 (Eurobio, Toulouse, France) supplemented with 10% heat-inactivated fetal bovine serum (FBS) (Eurobio), 2 mM L-glutamine, 100 U/ml penicillin, 100 µg/ml streptomycin, 0.01 M HEPES buffer and 1 mM sodium pyruvate (Eurobio). Cells were grown until 75% confluency and passaged using trypsin/ 1× EDTA (Eurobio), washed with 1× PBS and resuspended in supplemented RPMI-1640. Expanded ILC2s were cultured in RPMI-1640 (Gibco 61870-010) supplemented with 8% human serum, non-essential amino acids (1%, Gibco 11140-035), sodium pyruvate (1%, Gibco 11360-039), penicillin–streptomycin (1%, Gibco 15140-114), L-glutamine (1%, Gibco 25030-024), kanamycin (0.1 mg/mL, Gibco 15160-054), 2β-mercaptoethanol (0.1%, Sigma-Aldrich M7522), IL-2 (100 U/mL, Roche) and IL-7 (5 ng/ml, Peprotech). For ILC2 expansion, freshly-sorted ILC2s were cultured for 2 weeks in supplemented RPMI-1640 with a mixture of IL-2 (100 U/mL), IL-7 (5 ng/ml) and by adding PHA (1 µg/ml) at day 0. Medium was replaced every 2–3 days and cell phenotype was verified by flow cytometry. When indicated, IL-33 (50 ng/ml, Adipogen), IL-25 (50 ng/ml, Biolegend), T0070907 (2 µM, Cayman Chemical), Celecoxib (1 µM, Sigma-Aldrich), anti-human IL-13 Antibody (10 µg/ml, Biolegend) and HpARI (1 µg/ml, Adipogen) were added. In particular, ILC2s were pre-treated with T0070907 for 48 h and then incubated in the presence of IL-33 and IL-25, or SW1116 CRC cells for an additional 24 or 48 h prior performing cytokine production analysis by flow cytometry and Legendplex analysis, respectively. For the evaluation of cytokine production, cells were stimulated with 1 µg/mL PMA plus 0.5 µg/mL Ionomycin, in the presence of BrefeldinA (all from Sigma-Aldrich) for 3 h prior to intracellular staining.

CD4+ Th cell subsets were cultured for 2 weeks in supplemented with 100 U/ml of IL-2 for Th1 and Th2 subsets, or 20 U/ml of IL-2 for the Th17 subset.

Expanded mouse ILC2s were cultured in RPMI-1640 supplemented with 10% heat-inactivated FBS, IL-2 (100 U/mL) and IL-7 (5 ng/ml). Freshly-sorted ILC2s from PPARgflox/floxId2-CreERT2 mice were incubated in RPMI-1640 supplemented with 10% heat-inactivated FBS, IL-2 (100 U/mL), IL-7 (5 ng/ml), IL-33 (50 ng/ml) and IL-25 (50 ng/ml). After 48 h the supernatant was collected and analyzed to detect cytokine production by Legendplex analysis.

To generate MC-38-IL-33 cells, we first used lentivirus to produce a lentivirus encoding *Il33*, as described previously[75]. Briefly, HEK-293 cells were transfected with pMD2G, pMDLg/pRRE, pRSV-Rev, and pLV plasmids (Cyagen Biosciences, Santa Clara, CA) encoding EGFP with full-length mouse *Il33*. Next, lentiviruses were harvested to infect MC-38 cells. Transduced MC-38 cells were sort-purified based on EGFP expression, which were then kept under antibiotic selection using 6 µg/ml puromycin for 2 weeks. The procedure was repeated a second time to select the cells with the highest transgene expression.

SW1116 was purchased from ATCC; MC-38 colon adenocarcinoma cell line was ordered from Kerafast, Boston, Cat# ENH204 Lot# 011317 (provided by James W. Hodge, PhD, MBA, National Cancer Institute/NIH). All cell lines were periodically tested for mycoplasma contamination and confirmed negative by PCR analysis.

**Source of mRNA sequencing data**. The expression level of PPAR genes was assessed using previously published mRNA sequencing data of sorted human ILC and Th subsets (see Methods in Ercolano et al.[1], raw count data per gene and per sample deposited in the ArrayExpress under accession number E-MTAB-8494). The raw count data were processed and converted to log2 normalized counts per million as described in Ercolano et al.[1] mRNA sequencing data are deposited in the ArrayExpress under accession number E-MTAB-8494.

**Peroxisome proliferator response element (PPRE) motif search in gene promoters**. The promoter sequences (+100 bp up- and downstream flanking regions) of human or mouse IL-5 and IL-13 were obtained from the Eukaryotic Promoter Database (human genome version Dec 2013 GRCh38, mouse genome version Mar 2012 mm10; https://epd.epfl.ch//index.php). The consensus sequences of the binding motif of the human or mouse PPARγ-RXRα heterodimer were downloaded from the Jaspar2020 database (matrix IDs: MA0065.1 (human), MA0065.2 (mouse), http://jaspar.genereg.net/). Two methods were used to determine the presence of the PPRE. First, the findMotifs.pl method of the HOMER software[76] (v.4.11, http://homer.ucsd.edu/homer/) and the weight matrix of the most conserved region of the motif (TCAAAGG for human and CAAGG for mouse) were used, allowing to search for similar motives with up to 2 mismatches. Second, we used the online FIMO tool of the MEME suite (v.5.1.1, http://meme-suite.org/index.html) by providing the complete motif weight matrices and the promoters' fasta sequences.

**Quantitative real-time PCR (qPCR)**. Total RNA was isolated from highly pure, sorted human ILC subsets using the TRIZOL reagent according to the manufacturer's instructions (Invitrogen). Final preparation of RNA was considered DNA- and protein-free if the ratio of readings at 260/280 nm was ≥1.7. Isolated mRNA was reverse-transcribed by iScript Reverse Transcription Supermix for RT-qPCR (Bio-Rad, Milan, Italy). The quantitative real-time PCR was carried out in the Applied Biosystems 7900HT Fast Real-Time PCR Sequence Detection System (Applied Biosystems) with specific primers (hPPARA 5′-ACGATTCGACTCAAGCTGGT-3′, 5′-CGACAGAAAGGCACTTGTGA-3′; hPPARB 5′-GCATGAAGCTGGAGTAC-GAGAAG-3′, 5′-GCATCCGACCAAAACGGATA-3′; hPPARG 5′-AAGGCCATTT TCTCAAACGA-3′, 5′-AGGAGTGGGAGTGGTCTTCC-3′; hIL33 5′-GTGACGGT GTTGATGGTAAGAT-3′, 5′-AGCTCCACAGAGTGTTCCTTG-3′; hMMP9 5′-GG GACGCAGACATCGTCATC-3′, 5′-TCGTCATCGTCGAAATGGGC-3′; hNCAD 5′-TCAGGCGTCTGTAGAGGCTT-3′, 5′-ATGCACATCCTTCGATAAGACTG-3′; hCPT1A 5′-CCGTAGCTGACTCGGTACTC-3′, 5′-TCTAAGAGCTTCATGGCTC AG-3′; mPparg 5′-GTGATGGAAGACCACTCGCATT-3′, 5′-CCATGAGGGAGT TAGAAGGTTC-3′; mMmp9 5′-GCAGAGGCATACTTGTACCG-3′, 5′-TGATGT TATGATGGTCCCCACTTG-3′; mNcad 5′-CTCCAACGGGCATCTTCATTAT-3′, 5′-CAAGTGAAACCGGGCTATCAG-3′) using KAPA SYBR® FAST qPCR Kits (Roche). Samples were amplified simultaneously in triplicate in one-assay run with a non-template control blank for each primer pair to control for contamination or for primer dimerization, and the Ct value for each experimental group was determined. The housekeeping genes (ribosomal protein S16 and beta-2-microglobulin (β2M)) were used as an internal control to normalize the Ct values, using the $2^{-\Delta Ct}$ formula.

Alternatively, snap-frozen tumor and matched non-tumor tissue from CRC patients ($n = 10 + 10$) were homogenized using a TissueLyser (Qiagen) (3 × 30 s at 30 Hz). Total RNA was isolated following the protocol for TRI-reagent (Sigma-Aldrich) and eluted in 50 µl of water. Total RNA was then reverse-transcribed into cDNA using a Promega Kit containing M-MLV-RT and Oligo(dT) primers, following the manufacturer's protocol.

**Preparation of cellular extracts and western blot analysis**. Whole-cell extracts were prepared using RIPA buffer (50 mM Tris, 150 mM NaCl, 1% Triton X-100, 1% sodium deoxycholate, 0.1% sodium dodecyl sulfate (SDS), 1 mM phenylmethylsulfonyl fluoride (PMSF), 1 mM Na3VO4, 5 mM NaF, and 1% cocktail protease inhibitors; Sigma). The protein concentration was measured by the Pierce™ BCA Protein Assay Kit (Thermo Scientific). Equal amounts of protein (40 µg/sample) were separated by electrophoresis in a 12% denatured polyacrylamide gel and blotted onto nitrocellulose membranes (Biorad). The membranes were blocked for 1 h in 5% low-fat milk in 1× PBS with 0.1% Tween20 (PBST) at room temperature. The filters were then incubated overnight at +4 °C with the following primary antibodies: PPARγ (2443, Cell Signaling, diluted 1:1000) and α-Tubulin (3873, Cell Signaling, diluted 1:1000). The membranes were washed 3 times for 10 min with PBST and then incubated with HRP-conjugated anti-rabbit or anti-mouse antibodies (Biolegend) for 2 h at room temperature. The membranes were then washed for 3 times for 10 min in PBST, and proteins were visualized by the ECL chemiluminescence method. The immunoreactive bands of proteins were acquired using GBOX Chemi XX6 system (Syngene).

**PPARγ small interfering RNA transfection of ILC2s**. For the silencing experiments, expanded ILC2s were seeded in 96-well plates ($2 \times 10^4$ cell/well) and transfected the next day, according to the manufacturer's instruction, with PPARG Trilencer-27 Human siRNA (OriGene, Rockville, MD, USA) (rGrGrArArUrUrAr GrA rUrGrArCrArGrCrGrArCrUrGrUrGCA; rUrCrCrGrArGrArArArCrArArUr CrArGrArUrUrUrGrGrArAGCT; rArGrArArCrArArUrCrCrArGrUrGrGrUrUr GrCrArGrATT). The final concentration of the siRNA pool was 10 nM. 48 h after transfection, IL-13 and IL-5 production and secretion was evaluated by flow cytometry and Legendplex analysis, respectively. The knockdown of PPARγ expression in cells after transfection was confirmed by qPCR analysis. The Universal scrambled negative control siRNA duplex was used as negative control.

**ChIP assay**. Expanded human ILC2s were cross-linked with 1% formaldehyde for 15 min. Chromatin was sheared by sonication with Bioruptor Pico (30" on and

30″ off for 25 cycles) and immunoprecipitated with anti-PPARγ (5 μg, Diagenode) or an isotype control IgG using MAGnify Chromatin Immunoprecipitation system (Thermo Fisher). Eluted DNA was used for qPCR analysis using *IL-13* primers (5′-GAAGGCTCCGCTCTGCAAT-3′, 5′-TCCAGGGCTGCACAGTACA-3′).

**Electron microscopy (EM)**. Sorted cells were fixed in glutaraldehyde solution (EMS, Hatfield, PA, US) 2.5% in phosphate buffer (PB 0.1 M pH7.4) (Sigma, St Louis, MO, US) during 1 h at room temperature (RT). Then, they were directly post-fixed by a fresh mixture of osmium tetroxide 1% (EMS, Hatfield, PA, US) with 1.5% of potassium ferrocyanide (Sigma, St Louis, MO, US) in PB buffer during 1 h at RT. The samples were then washed three times in distilled water and dehydrated in acetone solution (Sigma, St Louis, MO, US) at graded concentrations (30%—40 min; 50%—40 min; 70%—40 min; 100%—3×1 h). This was followed by infiltration in Epon (Sigma, St Louis, MO, US) at graded concentrations (Epon 1/3 acetone-2h; Epon 3/1 acetone-2h, Epon 1/1-4 h; Epon 1/1-12 h) and finally polymerized for 48 h at 60 °C in the oven. Ultrathin sections of 50 nm were cut on a Leica Ultracut (Leica Mikrosysteme GmbH, Vienna, Austria) and picked up on a copper slot grid 2×1 mm (EMS, Hatfield, PA, US) coated with a polystyrene film (Sigma, St Louis, MO, US). Sections were post-stained with uranyl acetate (Sigma, St Louis, MO, US) 4% in $H_2O$ during 10 min, rinsed several times with $H_2O$ followed by Reynolds lead citrate in $H_2O$ (Sigma, St Louis, MO, US) during 10 min and rinsed several times with $H_2O$. Micrographs were taken at a magnification of 4800X with a transmission electron microscope Philips CM100 (Thermo Fisher Scientific, Waltham, MA USA) at an acceleration voltage of 80 kV with a TVIPS TemCam-F416 digital camera (TVIPS GmbH, Gauting, Germany) using EMMENU software and exported in 8bits. The acquisition and analysis of the images were performed double-blinded using 3dmod and its stereology plugin[77]. Briefly, a grid was applied on each cell and each intersection was defined as being part of the nucleus, cytoplasm or mitochondria, allowing to determine the percentage of mitochondrial volume per ILC2.

**Multiplex cytokine assay**. The concentrations of various cytokines in ILC2 supernatants, healthy donor's and patient's sera were determined using the multi-LEGENDplex™ analyte flow assay kit (human and mouse Th Panel (13-plex), Biolegend). Briefly, antibodies specific for the 13 analytes were conjugated to 13 different fluorescence-encoded beads. The beads were mixed with the supernatants, incubated for 2 h at room temperature, washed, and incubated for 1 h with detection antibodies. Finally, streptavidin-PE was added and incubated for 30 min, and the beads were washed and acquired using Gallios. The results were analyzed by using the Legendplex software (version 8.0).

**ELISA**. IL-33 plasma concentrations were evaluated using ELISA kit according to the manufacturer's instruction (LEGEND MAX™ Human IL-33 ELISA Kit, Biolegend).

**IHC**. Human CRC tissues were fixed in 4% formaldehyde and embedded in paraffin. Staining reactions were performed by automated staining using a BOND RX autostainer (Leica Biosystems). For double immunohistochemistry staining, sections were first deparaffinized and antigen was retrieved using 1 mM Tris solution (pH 9.0) for 30 min at 95 °C. Sections were then stained with goat anti-human IL-33 (R&D Systems, # AF3625; dilution of 1:400) and anti-human CD3 (abcam, # ab16669; dilution of 1:400) or anti-human IL-33 and anti-human GATA3 (Cell Marque, #390M-14; dilution of 1:400) primary antibodies. A rabbit anti-goat antibody (Agilent, # E0466) was used as secondary antibody, at a dilution of 1:400. Specific binding of primary antibodies was visualized using a polymer-based visualizing system with horseradish peroxidase as the enzyme and 3,3-diaminobenzidine (DAB) as a brown chromogen, or an alkaline phosphatase-linked polymer and Fast Red as red chromogen (Bond™ Polymer Refine and Red Detection), respectively (all from Leica Biosystems). The samples were counterstained with hematoxylin and mounted with Aquatex (Merck). Slides were scanned in high resolution on whole slide scanners Pannoramic 250 Flash (3DHISTECH) or NanoZoomer S360 (Hamamatsu). All human CRC tissues were provided by the Tissue Bank Bern.

**Image processing**. Each TMA image was dearrayed and individual cores exported as PNG using QuPath[78]. Color deconvolution was performed using ImageJ (Version 1.52p)[79] and it's "color deconvolution" plugin[80] resulting in three individual channels for hematoxylin, IL33 and CD3 or GATA3, respectively. Since the CD3 and GATA3 stainings were performed on serial section, they could be co-registered using the ImageJ plugin "bunwarpJ"[81] based on the hematoxylin channel.

**In vitro co-culture experiments**. Expanded ILC2s from healthy donors were pretreated with T0070907 PPARγ antagonist for 48 h and then cultured with the SW1116 colon cancer cell line in a ratio of 1:1. After 48 h, ILC2s were harvested and analyzed to detect cytokine production by flow cytometry analysis.

**Preparation of ILC2 conditioned medium (CM)**. Expanded ILC2s were pretreated or not with T0070907 PPARγ antagonist for 48 h and then stimulated with

IL-33 and IL-25 (50 ng/ml). After 48 h the medium was collected, centrifuged at $250 \times g$ for 5 min and filtered through a 0.22-μm syringe filter.

**Wound healing assay**. SW1116 cells were seeded in 24-well plates ($3 \times 10^5$ cells/well). Once the cells reached 90% confluence, ILC2 or ILC2 + T0070907 CM (30% v/v) were added and a wound area was carefully created by scraping the cell monolayer with a sterile 200 μl pipette tip. Subsequently, the cells were incubated at 37 °C in 5% $CO_2$. The width of the wounded area was monitored and photographed with an inverted microscope at various time points (20-fold magnification). The wounded area was measured using Image J software (LASV3.8, Germany).

**Growth rate analysis**. Cell growth rate was determined by MTT (3-(4,3-(4,5-dimethylthiazol-2-yl)-2,5 diphenyltetrazolium bromide 5-dimethylthiazol-2-yl)-2, 5-diphenyltetrazolium bromide) assay. Briefly, SW1116 cells were seeded in 96-well plates ($5 \times 10^3$ cells/well) and incubated for 24, 48, and 72 h with ILC2 or ILC2 + T0070907 CM (30% v/v) before adding 25 μL MTT (Sigma-Aldrich; 5 mg/mL in saline). Cells were then incubated for additional 3 h at 37 °C. After this time interval, dark blue crystals were solubilized with pure DMSO. The optical density of each well was measured with a microplate spectrophotometer (TitertekMultiskan MCC/340), equipped with a 540-nm filter.

**Clonogenic assay**. SW1116 cells ($1 \times 10^3$ cells/well) were seeded in 6-well plates with ILC2, ILC2 + T0070907 CM (30% v/v) or ILC2 CM with anti-IL-13 blocking antibody (IL13i). Cells were cultured for 14 days to allow the colonies to form. Formed colonies were washed twice with 1xPBS, fixed by 4% paraformaldehyde, and stained with 0.5% crystal violet and colonies containing more than 50 cells (established by microscopy) were counted manually. Images of the colonies were obtained using a digital camera. The experiments were done at least three times, in duplicate.

**In vivo tumor models**. PPARγ[flox/flox] mice[82] were backcrossed to C57BL/6[83] and crossed in house with Id2-CreER[T2] (Jackson Laboratory, Stock number 016222). To induce the deletion of PPARγ after intraperitoneal (i.p.) tamoxifen administration. RORα[fl/sg]Il7rCre mice were kindly provided by Prof. A. McKenzie[45]. PPARγ[flox/flox]Id2-CreER[T2] and RORα[fl/sg]Il7rCre mice and littermates, were between 6- and 12-weeks age and bred in house. All our mouse strains are in a C57BL/6 genetic background. $5 \times 10^5$ MC-38-IL33 murine colorectal cancer cells were injected subcutaneously (s.c.) in 200 μL of 1× PBS in the right flank and tumor growth was monitored. When indicated, T0070907 (7.5 mg/kg, Cayman Chemical), anti-IL13 neutralizing antibodies (50 μg/mouse, InvivoGen) and Isotype controls (50 μg/mouse, Bxcell) were daily injected intraperitoneally (i.p.) for 18 days. For ILC2 adoptive transfer, donor mice were injected with 0.4 μg of IL-25 and IL-33 for induction of ILC2s. 3 days after injection, flow cytometry-purified donor lung ILC2s were expanded in vitro (see Cell Culture) before $2.5 \times 10^5$ intravenous (i.v.) injection into RORα[fl/sg]Il7rCre mice. Tumor sizes were measured using a digital caliper and tumor volumes were calculated using the following equation: tumor volume = $II/6(D1 \times D2 \times D3)$ where $D1$ = length; $D2$ = width; $D3$ = height and expressed as $cm^3$. We have complied with all relevant ethical regulations for animal testing and research. This study was approved by the Veterinary Authority of the Swiss Canton Vaud (authorization no. VD3255.f) and performed in accordance with Swiss ethical guidelines. All animals were maintained at the University of Lausanne's animal facility under a 12 h dark/light cycle, at 21 °C ± 1 °C and 55% ± 10% HR.

**Statistical analysis**. Statistical analysis was performed using GraphPad Prism software version 6. For comparison of two groups the $t$ test was used, for comparison of multiple groups ANOVAs or the non-parametric (Mann–Whitney or Kruskal–Wallis tests) tests were used. The data are shown by plotting individual data points and the mean ± SEM. A $p$ value < 0.05 (two-tailed) was considered statistically significant and labeled with *$p$ values < 0.01, 0.001 or 0.0001 were labeled with **, *** or ****, respectively.

**Reporting summary**. Further information on research design is available in the Nature Research Reporting Summary linked to this article.

## Data availability

The RNA-seq data used in this study are deposited in the ArrayExpress under accession number E-MTAB-8494. Proteins putatively involved in protein–protein interactions were taken from the STRING database. The promoter sequences (+100 bp up- and downstream flanking regions) of human or mouse IL-5 and IL-13 were obtained from the Eukaryotic Promoter Database (human genome version Dec 2013 GRCh38, mouse genome version Mar 2012 mm10; https://epd.epfl.ch//index.php). The consensus sequences of the binding motif of the human or mouse PPARγ-RXRα heterodimer were downloaded from the Jaspar2020 database (matrix IDs: MA0065.1 (human), MA0065.2 (mouse), http://jaspar.genereg.net/). Source data are available as a Source data file. The remaining data are available within the Article, Supplementary Information or available from the authors upon request.

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

## Acknowledgements

We thank healthy donors and patients for the participation to this study. We thank Bérengère Salomé for the generation of the mRNA sequencing dataset and Anthony Cornu for the excellent technical assistance. We highly appreciated the assistance from the Electron Microscopy facility (EMF) at the University of Lausanne and particularly thank Jean Daraspe for his excellent support. We thank the animal facility and the flow cytometry facility at the University of Lausanne and University of Geneva for the excellent assistance. We thank Ioannis Kritikos for generating MC-38-IL-33 cells and Kristýna Hlaváčková for making cDNA from human CRC tissues, as well as the Translational Research Unit of the Institute of Pathology of the University of Bern—in particular Irene Centeno-Ramos, José Galván, and Inti Zlobec. This work was supported by grants from the Swiss National Science Foundation (PRIMA PR00P3_179727) to C.J., from the Swiss Cancer League (KFS-4402-02-2018) to C.J., from the Helmut Horten Foundation (to P.K. and C.J.) and from a Bourse Pro-Femmes, University of Lausanne to S.T. P.R was supported in part by grants from the Swiss National Science Foundation (FNS 31003A_156469 and FNS 310030_182735). P.C.O. was supported by grant from the Swiss National Science Foundation (FNS 31003A_182470). M.K.O. was supported by a grant from the Swiss National Science Foundation (FNS 310030B_182829). G.E. was supported by Fondazione Umberto Veronesi.

## Author contributions

G.E. performed the experiments, analyzed the data, and wrote the paper. A.G.C., G.V., N.D., and M.K.R. performed the experiment and analyzed the data. T.W. performed bioinformatics analysis. C.B., R.L., and P.K. provided CRC patient samples and revised the manuscript. L.M., A.I., P.C.O., M.K.O., D.M., and P.R. provided intellectual contributions and revised the manuscript. S.T. contributed to flow cytometry panel design, data analysis and provided intellectual contribution. C.J. provided intellectual contribution, supervised all the experiments, critically revised the manuscript and gave final approval to the publication.

## Competing interests

The authors declare no competing interests.
