## [Peer Review File · Nature Communications]

REVIEWER COMMENTS

Reviewer #1 (Remarks to the Author): with expertise in ILC and metabolism

The paper by Ercolano et al. investigates the importance of PPAR γ in driving pro-tumorigenic ILC2 responses. The authors show that human and mouse ILC2 express PPAR γ and that blockade or genetic deletion of PPAR γ in ILC2 impairs cytokine production of ILC2 and thus their pro-tumoral function. This suggests that PPAR γ could be a potential target in cancer immunotherapy. While the analysis performed is interesting some of the findings are based primarily on correlations, which allows alternative interpretations. In particular it remains unclear how PPAR γ drives cytokine production in ILC2. Although the authors mention that PPAR γ controls lipids metabolism, no experiments are included to address the involvement of this important metabolic pathway. This is in particular important since published data suggest an essential role of PPAR γ by promoting lipid metabolism in ILC2 in the context of airway inflammation (Karagiannis et al. 2020 Immunity). Hence, it would be important to provide new mechanistic insights of how PPAR γ drives the pro-tumoral function of ILC2 in vivo.

Specific points:

Fig.1: Lack of PPAR γ expression in T helper cell subsets stands in discrepancy to published reports showing the importance of PPAR γ in T cells activation. (Angela et al. 2016 Nat Comm). How do the authors explain this discrepancy?

It would be helpful to provide information on how human T helper cell subsets shown in Fig. S1 are identified and purified?

Fig. 2: Recent reports identified PPAR γ as an important factor driving mouse ILC2 proliferation and survival (Karagiannis et al. 2020 Immunity). Are human ILC2 likewise impaired upon T0070907 treatment?

Is PPAR γ directly controlling PD1 expression in ILC2 as suggested by (Batyruva et al. 2020 Immun Inflamm Dis.)?

Fig. 3: Mitotracker dyes and TMRM allow to assess the mitochondrial mass and membrane potential. However, they do not allow assessing mitochondrial respiration. More should be done to quantify more standard parameters of cellular metabolism (such as extracellular flux analysis or similar metabolic assay) in response to PPAR γ inhibition or deletion.

In Fig. 4 ILC2 numbers should be displayed as numbers of total lymphocytes and ideally as total cell counts. The heading of Fig. 4 implies that a direct connection between IL-33 and ILC2 is investigated although the data supplied is a mere correlation. Increase of IL-33 plasma levels do not appear to result in increased PPAR γ levels in ILC2. More should be done to assess the connection between tumor derived IL-33 and increased PPAR γ expression in vivo.

In this regard the authors could use blocking antibodies for IL33R or IL33 to confirm that the ILC2 tumor cross-talk is driven by IL-33 in vitro (Fig. 5a) and in vivo (Fig. 6).

What were the culture conditions for mouse ILC2 displayed in Fig 6? For how long were ILC2 from WT and ID2Cre PPAR γ fl/fl mice incubated?

Fig. 7: Do ID2Cre PPAR γ fl/fl mice also display a difference in survival?

Is PPAR γ upregulated in TIL ILC2 and how is PPAR γ activated in TIL ILC2? The authors should link IL-33 production to the expression of PPAR γ in ILC2. Do other ILC subtypes expressing ID2, such as ILC1/NK cells also express PPAR γ in the tumor microenvironment?

Could the PPAR γ inhibitor also be used therapeutically as suggested by the authors? Maybe the authors could include experiments to test the efficacy of this treatment.

The authors suggest that PPAR γ may be a relevant target for asthma and allergies. This work has been done and could be included in the discussion of the study (Chen et al. Science Immunol. 2017, Karagiannis et al. 2020 Immunity).

Reviewer #2 (Remarks to the Author): with expertise in colon cancer and IL33 signaling

The manuscript builds up on the previous discovery of ILC2s that play a critical role in protection against helminths and in diverse inflammatory diseases by responding to soluble factors such as the alarmin IL-33 through the receptor ST2. Here, the authors show that peroxisome proliferator-activated receptor gamma (PPAR γ) but not PPAR α and PPAR β is selectively expressed in human expanded ILC2s but not ILC1s and ILCPs. Selective PPAR γ antagonist T0070907 inhibition impaired IL-13 production and mitochondrial fitness. Using a COX-2 inhibitor, they showed a slight decrease in IL-13 secretion. Although supp fig 3 show no difference in PPAR γ expression in HDs vs. CRC patients, the authors indicated that PPAR γ blockade in ILC2s disrupted their pro-tumoral effect induced by IL-33-secreting cancer cells in human samples. They further confirmed this ILC2-driven protumoral effect by using Ppargfl/flId2CreERT2 mice and MC38 CRC model, and showed significantly suppressed tumor growth in vivo. The authors concluded that PPAR γ targeting might be exploited in cancer immunotherapy and in other ILC2-driven mediated disorders, such as asthma and allergy.

Overall, the data on PPAR γ in ILC2s are novel, and the data are well presented. However, we have major criticisms shown below:

Major:

- 1) The major concern is that Supp Figure 3 indicated no significant difference of the PPAR γ expression between freshly sorted ILC2s from healthy donor as compared to CRC patients, which is concerning for their hypothesis on tumor microenvironment. The point by point following comments derive from this major concern.
- 2) Figure 1 to Figure 3, we just see the ILCs data from healthy donors but the paper is about tumor microenvironment and data from tumor-infiltrated ILCs in patients or in murine models are needed.
- 3) Figure1 display the expression of PPAR γ on different human ILCs subsets at mRNA level. Verification at the protein level of PPAR γ by flow cytometry will strengthen the data. Also, Supp. Figure1 just show the expression of PPAR γ at mRNA level in CD4 T helper subsets, if the protein level of PPAR γ could be further verified by flow cytometry.
- 4) The data of binding motif in the promoter region of IL-5 and IL-13 was generated by the HOMER software (Figure2d and 6f). It will be preferred to verify this with cell lines by co-IP and any other biochemistry pathway.
- 5) MitoTracker Green and Deep Red dye uptake were significantly decreased in ILC2s after treatment with PPAR γ inhibitor, suggesting reduced mitochondrial mass and impaired respiration, respectively (Figure 3). However, if the authors would like to focus on the ILC2s-mediated immunotherapy, they should use tumor-infiltrated ILC2s from patients or murine models.
- 6) The data displayed in Figure 4a, 4b and 4c have previously been published by the Krebs' group.
- 7) It would be important to show the effect of IL-13 in the in vivo data with for example with IL13 knockout or anti-IL-13 tumor model.
- 8) In Figure 6, could the authors explain why they sorted the ILC2s from lung tissues instead of the intestinal tissues, while the authors used a CRC model?

Minor:

- 1) The authors should provide high resolution graph of Figure 2d and 6f.
- 2) As a represent flow data (Figure 2a, 4d and 6b), the percentage should be labeled in the graph.
- 3) The tumor volume in Figure 7c should be monitored for at least 28 days.

Reviewer #3 (Remarks to the Author): with expertise in ILC

Ercolano et al. examined the role of PPAR γ for the pro-tumor activity of group 2 innate lymphoid cells (ILC2) in both human and mouse. The authors showed the inhibitor of PPAR γ , T0070907 suppressed IL-5 and IL-13 secretion from both human and mouse ILC2. PPAR γ inhibition was shown to suppress mitochondrial function as well. Importance of PPAR γ in IL-5 and IL-13 secretion was also confirmed by deletion of PPAR γ in mouse. The authors then examined the pro-tumor effect of ILC2 using colorectal cancer (CRC) in both human and mouse. The authors showed that ILC2-derived IL-13 stimulated migration of SW1116 human CRC cell line associated with the induction of EMT markers MMP9 and N-cadherin. The authors also showed the presence of IL-33 and CD3-GATA3⁺ ILC2 in CRC tissues. In a mouse model, IL-33 expressing MC38 mouse CRC cell line was better controlled in PPAR γ -deficient mice. Mouse deficient for ILC2 survived longer with MC38-IL-33 compared with control mice. From these results, the authors propose that PPAR γ is a potential target for cancer immunotherapy.

This study was well designed and performed in a logical manner to provide readers with some new findings. As for the role of PPAR γ in ILC2 functions, Karagiannis et al. have already reported that PPAR γ together with Dgat1 and mTOR controls lipid metabolism in ILC2 (PMID: 32268121). However, this paper was not even mentioned in the manuscript, which is not appropriate. One of major potential advance of Ercolano et al. paper over Karagiannis et al. papers is that Ercolano et al. propose that PPAR γ directly activates IL-5 and IL-13 gene transcription with RARA but no evidence is provided, which dampens the priority of this paper.

Although this paper provides the readers with some new findings, there are several points that need to be clarified.

- 1) To confirm the pro-tumor effect of ILC2 in Figures 7a and 7b, the authors should adoptively transfer wild type ILC2 into ROR α ^{<fl/sg>}IL7rCre and Pparg^{<fl/fl>}Id2Cre⁺ mice and examine whether the transfer of ILC2 promote tumor progression and mortality.
- 2) Figures 2d and 6f: the authors must examine whether the motives are functional or not using either reporter assay in vitro or CRSPR-mediated gene deletion assay in vivo.
- 3) In figure 5g, the authors should also examine the effect of anti-IL-13 antibody as well.
- 4) Bottom of page8: because there is no significant difference in IL-5 and IL-13 secretion between T0070907 treatment and T0070907 plus CXB (Supplementary Figure 2b), the authors cannot conclude the interaction of COX-2 and PPAR γ .
- 5) As mentioned by the authors, the role of ILC2 on tumor immunity is multiphasic. Some papers reported anti-tumor activity of ILC2 against melanoma (PMID: 22174445, 32101749), suggesting that the benefit of suppressing ILC2 activities depends on the tumor type. The authors should discuss these points.
- 6) The authors should provide readers with detailed methods to obtain Th1, Th2, and Th17 cells ex vivo and in vitro expansion.
- 7) Similarly, methods for pre-treatment of ILC2 with T0070907 should be provided.
- 8) Genetic background of mice should be provided.
- 9) Stock number should be provided for mouse line obtained from the Jackson Laboratory.

10) The first line of page 2: "Type 2 innate lymphoid cells" should be "Group 2 innate lymphoid cells".

REVIEWERS' COMMENTS

Reviewer #1

The paper by Ercolano et al. investigates the importance of PPAR γ in driving pro-tumorigenic ILC2 responses. The authors show that human and mouse ILC2 express PPAR γ and that blockade or genetic deletion of PPAR γ in ILC2 impairs cytokine production of ILC2 and thus their pro-tumoral function. This suggests that PPAR γ could be a potential target in cancer immunotherapy. While the analysis performed is interesting some of the findings are based primarily on correlations, which allows alternative interpretations. In particular it remains unclear how PPAR γ drives cytokine production in ILC2. Although the authors mention that PPAR γ controls lipids metabolism, no experiments are included to address the involvement of this important metabolic pathway. This is in particular important since published data suggest an essential role of PPAR γ by promoting lipid metabolism in ILC2 in the context of airway inflammation (Karagiannis et al. 2020 Immunity). Hence, it would be important to provide new mechanistic insights of how PPAR γ drives the pro-tumoral function of ILC2 in vivo.

Specific points:

Fig.1: Lack of PPAR γ expression in T helper cell subsets stands in discrepancy to published reports showing the importance of PPAR γ in T cells activation. (Angela et al. 2016 Nat Comm). How do the authors explain this discrepancy?

***Answer:** We thank the reviewer for pointing out this important aspect. In the work by Angela et al., the authors showed that PPAR γ expression is increased in TCR/CD28 activated CD4 T cells. In contrast, in our work human Th2 cells express PPAR γ and that absence of PPAR γ in Th2 cells, but not in Th1 or Th17 cells affects their polarization. In the current manuscript, we analysed PPAR γ expression in freshly-sorted and in vitro expanded Th subsets (cultured in the presence of IL-2), without any TCR triggering. This differences in the experimental setups might explain the discrepant results with other reports.*

It would be helpful to provide information on how human T helper cell subsets shown in Fig. S1 are identified and purified?

***Answer:** As asked by the reviewer, Th subset identification strategy and culture conditions have been added in the Materials and methods section (pages 29,31).*

Fig. 2: Recent reports identified PPAR γ as an important factor driving mouse ILC2 proliferation and survival (Karagiannis et al. 2020 Immunity). Are human ILC2 likewise impaired upon T0070907 treatment?

***Answer:** We thank the reviewer for raising this point. We performed an MTT assay on human ILC2s after 72h treatment with T0070907. As shown in Figure 1 below, the PPAR γ antagonist did not impair human ILC2 proliferation.*

Figure 1: Proliferation rate was assessed by MTT assay in human ILC2s, after incubation with T0070907 for 72 hours.

Is PPARg directly controlling PD1 expression in ILC2 as suggested by (Batyruva et al. 2020 Immun Inflamm Dis.)?

Answer: We thank the reviewer for this comment related to the recent work by Batyruva et al. We assessed PD1 expression on mouse ILC2s, sorted from the lungs of $Pparg^{fl/fl} Id2Cre^+$ and $Pparg^{fl/fl} Id2Cre^-$ mice, after *in vitro* treatment with 4-Hydroxytamoxifen. As shown in Figure 2 below (Panel A), PD1 MFI is significantly reduced in PPARγ deficient ILC2s compared to WT ILC2s.

In addition, we also evaluated the expression of PD1 in human ILC2s after transfection with a PPARγ silencing RNA. Similar to the mouse results, human ILC2s also show a significant reduction of PD1 after PPARγ silencing (Panel B).

Figure 2: A, Frequency (left) and MFI (right) of PD1 expression in mouse expanded ILC2s from the lung of $Pparg^{fl/fl} Id2Cre^+$ mice, treated or not with 4-Hydroxytamoxifen. B, Frequency of PD1 in human expanded ILC2s after 48h silencing with a PPARG SiRNA.

Fig. 3: Mitotracker dyes and TMRM allow to assess the mitochondrial mass and membrane potential. However, they do not allow assessing mitochondrial respiration. More should be done to quantify more standard parameters of cellular metabolism (such as extracellular flux analysis or similar metabolic assay) in response to PPARg inhibition or deletion.

Answer: we thank the reviewer for this comment. Given the low number of human ILC2s, we were unable to perform Seahorse analysis with enough replicates to ensure statistical power. To address at best the comment raised by the reviewer on PPAR γ -mediated regulation of ILC2-mitochondrial fitness we used another experimental approach to evaluate mitochondrial fitness, which has been used in previous publications (Yi-Ru Yu et al., Nat Immunol. 2020; Zorova et al., Anal. Biochem. 2018). As reported in Figure 3 here below, we evaluated the changes of fluorescent intensity of Mitotracker Deep Red (MDR) in untreated vs T0070907 treated ILC2s, upon addition of Oligomycin. While in untreated cells the addition of Oligomycin resulted in a robust increase of MDR, this was not observed in T0070907 treated ILC2s, further supporting our conclusion that PPAR γ signal regulates mitochondrial activity in ILC2.

Figure 3: Quantification of Mitotracker Deep Red in human expanded ILC2s pre-treated or not with T0070907 and stimulated with Oligomycin for 2.5 hours. The result is represented as fold change MDR post- vs pre-Oligomycin.

In Fig. 4 ILC2 numbers should be displayed as numbers of total lymphocytes and ideally as total cell counts.

Answer: as suggested, we quantified ILC numbers as total cell counts. As shown in the Revised Figure 4d, the absolute numbers are comparable to ILC frequencies and no significant difference is observed in the different conditions.

The heading of Fig. 4 implies that a direct connection between IL-33 and ILC2 is investigated although the data supplied is a mere correlation. Increase of IL-33 plasma levels do not appear to result in increased PPAR γ levels in ILC2. More should be done to assess the connection between tumor derived IL-33 and increased PPAR γ expression in vivo. In this regard the authors could use blocking antibodies for IL33R or IL33 to confirm that the ILC2 tumor cross-talk is driven by IL-33 in vitro (Fig. 5a) and in vivo (Fig. 6).

Answer: We thank the reviewer for the suggestion. To gain further insight in the correlation between IL-33-derived from CRC cells and ILC2 function, we performed co-culture experiments using HpARI which is known to suppress type 2 immune responses through interference with the IL-33 pathway (Osburn et al., Immunity 2017). As shown in Figure 4 below (Panel A), the addition of HpARI in the medium significantly reduced the production of IL-13 by ILC2s, after co-culture with the SW1116 CRC cells. Similarly, co-culture experiments of murine ILC2s with the parental CRC MC38 or the MC38-IL33 cell line resulted in higher production of IL-13 by ILC2s cultured with MC38-IL33, as compared to MC38 wildtype cells. Panel A of this Figure has been added as revised Supplementary Figure 4a (pages 16, 18 and 30).

Figure 4: A, Frequencies of IL-13 positive ILC2s upon co-culture with SW1116 CRC cells, in the presence or absence of HpARI. B, Frequencies of IL-13 and IL-5 positive mouse ILC2s upon co-culture with the parental MC38 or MC38-IL33 CRC cells.

What were the culture conditions for mouse ILC2 displayed in Fig 6? For how long were ILC2 from WT and ID2Cre PPAR γ fl/fl mice incubated?

Fig. 7: Do ID2Cre PPAR γ fl/fl mice also display a difference in survival?

Answer: We thank the reviewer for raising these points. We included in the updated Materials and Methods section detailed information about mouse ILC2 culture conditions (page 31).

In addition, as requested, we also evaluated the survival of PPAR γ fl/fl mice after tumor injection. As shown in the revised Figure 7d, overall survival was significantly extended in PPAR γ fl/fl Id2 Cre⁺ mice compared to control animals (pages 22-23).

Is PPAR γ upregulated in TIL ILC2 and how is PPAR γ activated in TIL ILC2? The authors should link IL-33 production to the expression of PPAR γ in ILC2. Do other ILC subtypes expressing ID2, such as ILC1/NK cells also express PPAR γ in the tumor microenvironment?

Answer: We thank the reviewer for this comment. Following his/her suggestion, we evaluated the expression of PPAR γ in freshly-sorted ILCs and NKs from both PBMCs and TILs of CRC patients. We observed a trend for increased expression of PPAR γ in ILC2s in patients compared to HDs, whilst the expression of PPAR γ was low and comparable in ILC1s, ILC3s and NKs of HDs and patients (Figure 5 below, Panel A). Moreover, we also assessed the expression of CPT1A, one of the well-characterized, canonical PPAR γ -direct target genes involved in fatty acid oxidation (Mascaró et al., J Biol Chem. 1998). In line with the increased expression of PPAR γ , a trend for increased CPT1A expression was found in ILC2s from CRC patients compared to HDs which reflects an increase in PPAR γ activity (Panel B). The results on the expression of PPAR γ and CPT1A in ILC2s from CRC patients have been included in the revised Supplementary Figure 3, panels a-b (pages 13, 15, 32).

Figure 5: A, Expression of PPAR γ assessed by qPCR in human freshly sorted ILC subsets and NK cells from PBMCs of HDs, and PBMCs and TILs of CRC patients. B, Expression of CPT1A assessed by qPCR in freshly sorted ILC2s from HDs and CRC patients.

Could the PPAR γ inhibitor also be used therapeutically as suggested by the authors? Maybe the authors could include experiments to test the efficacy of this treatment.

Answer: *We thank the reviewer for raising this important translational aspect. To test the in vivo efficacy of T0070907, we implanted MC38-IL33 CRC cells in C57BL6 mice and treated the animals daily with T0070907 i.p. 7,5 mg/kg, as previously reported (Burton et al., PPAR Res. 2008). As shown in the revised supplementary Figure 6b we observed a significant reduction of both tumor volume and weight in T0070907 treated mice compared to controls.*

The authors suggest that PPAR γ may be a relevant target for asthma and allergies. This work has been done and could be included in the discussion of the study (Chen et al. Science Immunol. 2017, Karagiannis et al. 2020 Immunity).

Answer: *Answer: We thank the reviewer for this comment. As suggested, we included in the discussion the studies by Chen and Karagiannis, as well as our work (Nobs et al, JEM, 2017) (page 28).*

Reviewer #2

The manuscript builds up on the previous discovery of ILC2s that play a critical role in protection against helminths and in diverse inflammatory diseases by responding to soluble factors such as the alarmin IL-33 through the receptor ST2. Here, the authors show that peroxisome proliferator-activated receptor gamma (PPAR γ) but not PPAR α and PPAR β is selectively expressed in human expanded ILC2s but not ILC1s and ILCPs. Selective PPAR γ antagonist T0070907 inhibition impaired IL-13 production and mitochondrial fitness. Using a COX-2 inhibitor, they showed a slight decrease in IL-13 secretion. Although supp fig 3 show no difference in PPAR γ expression in HDs vs. CRC patients, the authors indicated that PPAR γ blockade in ILC2s disrupted their pro-tumoral effect induced by IL-33-secreting cancer cells in human samples. They further confirmed this ILC2-driven protumoral effect by using Ppargfl/flId2CreERT2 mice and MC38 CRC model, and showed significantly suppressed tumor growth in vivo. The authors concluded that PPAR γ targeting might be exploited in cancer immunotherapy and in other ILC2-driven mediated disorders, such as asthma and allergy.

Overall, the data on PPAR γ in ILC2s are novel, and the data are well presented. However, we have major criticisms shown below:

Major:

1) The major concern is that Supp Figure 3 indicated no significant difference of the PPAR γ expression between freshly sorted ILC2s from healthy donor as compared to CRC patients, which is concerning for their hypothesis on tumor microenvironment. The point by point following comments derive from this major concern.

Answer: We thank the reviewer for this comment, also raised by reviewer 1 (see Figure 6 of this point-by-point reply). As suggested, we increased the number of CRC patients to evaluate the expression of PPAR γ in freshly sorted ILC2s, both from PBMCs and TILs. As shown in Figure 6 of this point-by-point reply and in the revised Supplementary Figure 3 a-b, we observed a trend for increased expression of PPAR γ in ILC2s in patients compared to HDs, whilst the expression of PPAR γ was low and comparable in ILC1s, ILC3s and NKs of HDs and patients (Figure 6 PBP reply, Panel A). Moreover, we also assessed the expression of CPT1A, one of the well-characterized, canonical PPAR γ -direct target genes involved in fatty acid oxidation (Mascaró et al., J Biol Chem. 1998). In line with the increased expression of PPAR γ , a trend for increased CPT1A expression was found in ILC2s from CRC patients compared to HDs which reflects an increase in PPAR γ activity (Figure 6 PBP reply, Panel B). The results on the expression of PPAR γ and CPT1A in ILC2s from CRC patients have been included in the revised Supplementary Figure 3, panels a-b (pages 13, 15, 32).

2) Figure 1 to Figure 3, we just see the ILCs data from healthy donors but the paper is about tumor microenvironment and data from tumor-infiltrated ILCs in patients or in murine models are needed.

Answer: We agree with the reviewer that patients' data were lacking in our original manuscript. As suggested, we performed *in vitro* experiments using ILC2s from CRC patients. The inhibitory effects of T0070907 on cytokine production, cytokine secretion and mitochondrial fitness were confirmed in patients' ILC2s, as shown in the revised Supplementary Figure 3 c-e (pages 13, 15).

3) Figure 1 display the expression of PPAR γ on different human ILCs subsets at mRNA level. Verification at the protein level of PPAR γ by flow cytometry will strengthen the data. Also, Supp. Figure 1 just show the expression of PPAR γ at mRNA level in CD4 T helper subsets, if the protein level of PPAR γ could be further verified by flow cytometry.

Answer: We thank the reviewer for this comment. We tested several PPAR γ antibodies in flow cytometry, but in our hands, we were unable to obtain reliable results. However, we have previously shown (Nobs et al, JEM, 2017) that activated human Th2 cells express PPAR γ and that absence of PPAR γ in Th2 cells, but not in Th1 or Th17 cells affects their polarization. In the current manuscript, we analysed PPAR γ expression in freshly-sorted and *in vitro* expanded Th subsets (cultured in the presence of IL-2), without any TCR triggering.

4) The data of binding motif in the promoter region of IL-5 and IL-13 was generated by the HOMER software (Figure 2d and 6f). It will be preferred to verify this with cell lines by co-IP and any other biochemistry pathway.

Answer: We thank the reviewer for this suggestion. We performed chromatin immunoprecipitation (ChIP) on expanded human ILC2s stimulated or not with the cytokine cocktail composed of IL-33 and IL-25, and

treated or not with the PPAR γ antagonist T0070907. Our results, presented in the revised Supplementary Figure 2f, suggest that PPAR γ directly regulates IL-13 production during ILC2 activation (pages 8, 10, 34).

Regarding the murine PPAR γ , we were unable to perform chromatin immunoprecipitation (ChIP) on mouse ILC2s. However, we took advantage of the strategy used by Fali et al. (Mucosal Immunol. 2020) to evaluate the presence of PPAR γ binding sites in the vicinity of the *Il1rl1* gene, and analysed chromatin immunoprecipitation and sequencing (ChIP-Seq) data, available on the GEO (GSE115505) and EMBL-EBI (E-MTAB-7258) repositories. We searched for PPAR γ binding sites in the vicinity of the *IL-13* and *IL-5* genes, in mouse macrophages and in T helper cells polarized by IL-4 (Figure 6A and B here below). The results show the presence of several PPAR γ binding sites in Th2 cells, suggesting that PPAR γ might bind to these elements also in mouse ILC2s.

Figure 6: Data was downloaded from GEO:GSE115505 (Daniel et al., Immunity 2018) and EMBL-EBI:E-MTAB-7258 (Henriksson et al., Cell 2019) to search for bindings sites of PPAR γ in the IL-13 (A) and IL-5 (B) genes, as determined by sequencing of the chromatin immunoprecipitated with PPAR γ antibody upon IL-4-induced M2 and Th2 polarization.

5) MitoTracker Green and Deep Red dye uptake were significantly decreased in ILC2s after treatment with PPAR γ inhibitor, suggesting reduced mitochondrial mass and impaired respiration, respectively (Figure 3). However, if the authors would like to focus on the ILC2s-mediated immunotherapy, they should use tumor-infiltrated ILC2s from patients or murine models.

Answer: we thank the reviewer for the comment. As reported above in a similar comment raised by reviewer 1, we have now used expanded ILC2s from CRC patients to perform analyses of cytokine production, cytokine secretion and the mitochondrial assays. As shown in the revised Supplementary Figure 3 c-e, the inhibitory effects of T0070907 have been confirmed also in patients' ILC2s (pages 13, 15).

6) The data displayed in Figure 4a, 4b and 4c have previously been published by the Krebs' group.

Answer: while we agree with the reviewer that previous publications by the Krebs' group and by others reported increased IL-33 levels in CRC patients, these observations were not linked to ILC2 infiltration. In the current Figure 4, we have performed staining of IL-33 and of markers for ILC2 visualization (GATA3, CD3) on independent tissue sections as compared to previously published reports.

7) It would be important to show the effect of IL-13 in the in vivo data with for example with IL13 knockout or anti-IL-13 tumor model.

Answer: We thank the reviewer for the suggestion. To evaluate the role of IL-13 in CRC progression, we treated C57BL6 tumor-bearing mice with an anti-IL-13 neutralizing antibody, daily i.p. 50 µg/mouse, for eighteen days after tumor implant. As shown in the revised Supplementary Figure 6c, the neutralization of IL-13 significantly reduced both tumor volume and weight (pages 22, 24, 37).

8) In Figure 6, could the authors explain why they sorted the ILC2s from lung tissues instead of the intestinal tissues, while the authors used a CRC model?

Answer: We thank the reviewer for the comment. We used lung ILC2s since this organ is an easily accessible tissue and a high source of ILC2s. We also evaluated the expression of PPAR γ in the intestine of C57BL6 mice and could confirm that, also in this organ, PPAR γ is specifically highly expressed in ILC2s, while absent in ILC1s and ILC3s (Figure 7 here below).

Figure 7: Expression of Pparg assessed by qPCR analysis in freshly-sorted ILC subsets from the intestine of C57BL6 mice.

Minor:

1) The authors should provide high resolution graph of Figure 2d and 6f.

Answer: as requested by the reviewer, we increased the size of graphs in Figure 2d and 6f.

2) As a represent flow data (Figure 2a, 4d and 6b), the percentage should be labelled in the graph.

***Answer:** as suggested by the reviewer, we included the percentage of positive cells in the representative flow graphs.*

3) The tumor volume in Figure 7c should be monitored for at least 28 days.

***Answer:** we thank the reviewer for this comment. Unfortunately, we are unable to monitor tumor volumes 28 days after implant, since the volume would be higher than 1000 cc, the maximal limit set by the Swiss Veterinary Authorities.*

Reviewer #3

Ercolano et al. examined the role of PPAR γ for the pro-tumor activity of group 2 innate lymphoid cells (ILC2) in both human and mouse. The authors showed the inhibitor of PPAR γ , T0070907 suppressed IL-5 and IL-13 secretion from both human and mouse ILC2. PPAR γ inhibition was shown to suppress mitochondrial function as well. Importance of PPAR γ in IL-5 and IL-13 secretion was also confirmed by deletion of PPAR γ in mouse. The authors then examined the pro-tumor effect of ILC2 using colorectal cancer (CRC) in both human and mouse. The authors showed that ILC2-derived IL-13 stimulated migration of SW1116 human CRC cell line associated with the induction of EMT markers MMP9 and N-cadherin. The authors also showed the presence of IL-33 and CD3-GATA3⁺ ILC2 in CRC tissues. In a mouse model, IL-33 expressing MC38 mouse CRC cell line was better controlled in PPAR γ -deficient mice. Mouse deficient for ILC2 survived longer with MC38-IL-33 compared with control mice. From these results, the authors propose that PPAR γ is a potential target for cancer immunotherapy. This study was well designed and preformed in a logical manner to provide readers with some new findings. As for the role of PPAR γ in ILC2 functions, Karagiannis et al. have already reported that PPAR γ together with Dgat1 and mTOR controls lipid metabolism in ILC2 (PMID: 32268121). However, this paper was not even mentioned in the manuscript, which is not appropriate. One of major potential advance of Ercolano et al. paper over Karagiannis et al. papers is that Ercolano et al. propose that PPAR γ directly activates IL-5 and IL-13 gene transcription with RARA but no evidence is provided, which dampens the priority of this paper.

Although this paper provides the readers with some new findings, there are several points that need to be clarified.

1) To confirm the pro-tumor effect of ILC2 in Figures 7a and 7b, the authors should adoptively transfer wild type ILC2 into RORa^{<fl/sg>}IL7rCre and Pparg^{<fl/fl>}Id2Cre⁺ mice and examine whether the transfer of ILC2 promote tumor progression and mortality.

***Answer:** We thank the reviewer for the comment. As suggested, we performed ILC2 transfer from C57BL6 mice to ILC2 KO mice after implant of MC38-IL33 CRC cells. As shown in the revised Supplementary Figure 6a, ILC2 transfer in ILC2 KO mice significantly increased tumor growth compared to non-transferred ILC2 KO mice (pages 22, 24, 37).*

2) Figures 2d and 6f: the authors must examine whether the motives are functional or not using either reporter assay in vitro or CRSPR-mediated gene deletion assay in vivo.

***Answer:** we thank the reviewer for the comment. To evaluate if the motives identified are functional, we used different approaches. First of all, we transfected human expanded ILC2s with a SiRNA targeting PPAR γ and we evaluated cytokine production and secretion. We confirmed the reduction of IL-13 after PPAR γ silencing (Revised Supplementary Figure 2b-e). Moreover, we performed chromatin*

immunoprecipitation (ChIP) on expanded human ILC2s stimulated or not with the cytokine cocktail composed of IL-33 and IL-25, and treated or not with the PPAR γ antagonist T0070907. As shown in the revised Supplementary Figure 2f, our results suggest that PPAR γ directly regulates IL-13 production during ILC2 activation (pages 8, 10, 33-34).

Lastly, we also used YRS reporter mice recently described by Ricardo-Gonzalez et al., Nat Immunol. 2018. ILC2s were sorted from the lungs of these mice and stimulated in vitro with IL-33 and IL-25, in the presence or absence of T0070907. The addition of T0070907 reduced the production of both IL-13 and IL-5, as assessed using the reporter system (Figure 8 here below).

Figure 8: (Left) Representative example of flow cytometry analysis of mouse ILC2-derived IL-13 upon in vitro T0070907 treatment in YRS reporter mice. (Right) Frequencies of IL-13 and IL-5 positive ILC2s after T0070907 treatment in YRS reporter mice.

3) In figure 5g, the authors should also examine the effect of anti-IL-13 antibody as well.

Answer: *We thank the reviewer for the suggestion. As reported in the revised Figure 5g, we examined the expression of MMP9 and NCAD in SW1116 after incubation with ILC2 CM in the presence of an anti-IL-13 blocking antibody. We observed that the addition of the antibody reduced the EMT marker expression towards basal levels (pages 16-17).*

4) Bottom of page8: because there is no significant difference in IL-5 and IL-13 secretion between T0070907 treatment and T0070907 plus CXB (Supplementary Figure 2b), the authors cannot conclude the interaction of COX-2 and PPAR γ .

Answer: *we agree with the reviewer and have removed the speculation on a possible interaction of COX-2 and PPAR γ (page 9).*

5) As mentioned by the authors, the role of ILC2 on tumor immunity is multiphasic. Some papers reported anti-tumor activity of ILC2 against melanoma (PMID: 22174445, 32101749), suggesting that the benefit of suppressing ILC2 activities depends on the tumor type. The authors should discuss these points.

Answer: *we agree with the reviewer and have included this point in the discussion (page 26).*

6) The authors should provide readers with detailed methods to obtain Th1, Th2, and Th17 cells ex vivo and in vitro expansion.

Answer: as requested, details about Th subset sorting and culture have been added in the Materials and Methods section (pages 29,31).

7) Similarly, methods for pre-treatment of ILC2 with T0070907 should be provided.

Answer: as requested, details about T0070907 treatment have been added in the Materials and Methods section (page 30).

8) Genetic background of mice should be provided.

Answer: as requested, the genetic background of mice has been included in the Materials and Methods section (page 37).

9) Stock number should be provided for mouse line obtained from the Jackson Laboratory.

Answer: as requested, the stock number for the mouse obtained from the Jackson Laboratory has been included in the Materials and Methods section (page 37).

10) The first line of page 2: “Type 2 innate lymphoid cells” should be “Group 2 innate lymphoid cells”.

Answer: as requested, we have edited the text according to the suggestion (page 2).

REVIEWERS' COMMENTS

Reviewer #1 (Remarks to the Author):

I thank the authors for addressing my comments.

While this study was under review, two additional papers were published highlighting the importance of PPAR γ for the maintenance of lipid metabolism in ILC2 (Xia et al. PMID: 32811992, Fali et al. PMID: 33106586). These studies need to be included and the role of PPAR γ controlling lipid metabolism in ILC2 should be included in the discussion.

If I am not mistaken, the findings by Batyrva et al. are not included in the revised manuscript. This study should be added and discussed.

The authors observe no difference between untreated and T0070907 treated ILC2 when performing an MTT assay. Doesn't this result stand in direct contradiction to a reduction in mitochondrial function (assessed by TMRM/Mitotracker) upon PPAR γ inhibition?

As pointed out previously, Mitotracker dyes and TMRM allow to assess the mitochondrial mass and membrane potential as proxy for mitochondrial fitness. However, they do not allow assessing mitochondrial respiration. This needs to be clarified in the manuscript.

Reviewer #2 (Remarks to the Author):

We are pleased to see that all our concerns have been carefully addressed. Particularly some major issues have been substantially revised in the current manuscript. We are confident that this work will be of interest to the Nature communications readership.

Reviewer #3 (Remarks to the Author):

Ercolano et al. revised their paper with additional experimental results, which significantly improved the quality of the paper. There are only a few minor points that need to be addressed.

1) On page 13, the 11th and 12th lines from the bottom: although the authors stated that ILC2s from CRC patients' PBMCs produced more IL-13 and IL-5 compared to ILC2s from HDs' PBMCs, there is no significant difference for IL-5 (Fig. 4e & f). The authors should delete "and IL-5" from the sentence.

2) It is better to add results on the in vivo tumor growth in ILC2 KO mice as a part of Fig. 7a.

3) On page 27, the 6th to 8th lines from the bottom, the authors added a new sentence for the tumor-type dependent benefit of targeting ILC2s. A paper (PMID: 32101749) demonstrating the positive correlation between IL-33 expression and survival of melanoma patients should be cited here to show an example.

REVIEWERS' COMMENTS

Reviewer #1 (Remarks to the Author):

I thank the authors for addressing my comments.

While this study was under review, two additional papers were published highlighting the importance of PPAR γ for the maintenance of lipid metabolism in ILC2 (Xia et al. PMID: 32811992, Fali et al. PMID: 33106586). These studies need to be included and the role of PPAR γ controlling lipid metabolism in ILC2 should be included in the discussion.

Answer to the reviewer: We have included and discussed the studies by Xia et al., and Fali et al. (pages 19).

If I am not mistaken, the findings by Batyrva et al. are not included in the revised manuscript. This study should be added and discussed.

Answer to the reviewer: We have included and discussed the study from Batyrova et al. (page 19).

The authors observe no difference between untreated and T0070907 treated ILC2 when performing an MTT assay. Doesn't this result stand in direct contradiction to a reduction in mitochondrial function (assessed by TMRM/Mitotracker) upon PPAR γ inhibition?

Answer to the reviewer: Although the properties of MTT assay have been mainly associated to its mitochondrial metabolism, different data reported that only a minor fraction of MTT-formazan is deposited in mitochondria while most of the MTT-formazan is produced at large distance from the mitochondria (11933013) suggesting the involvement of other compartments such as endosomal vesicles (9231715). This could explain why we didn't observe any significant difference in the proliferation of ILC2 by using this experimental approach.

As pointed out previously, Mitotracker dyes and TMRM allow to assess the mitochondrial mass and membrane potential as proxy for mitochondrial fitness. However, they do not allow assessing mitochondrial respiration. This needs to be clarified in the manuscript.

Answer to the reviewer: We have clarified this comment in the manuscript.

Reviewer #2 (Remarks to the Author):

We are pleased to see that all our concerns have been carefully addressed. Particularly some major issues have been substantially revised in the current manuscript. We are confident that this work will be of interest to the Nature communications readership.

Answer to the reviewer: We thank the reviewer for the positive evaluation of our revised manuscript.

Reviewer #3 (Remarks to the Author):

Ercolano et al. revised their paper with additional experimental results, which significantly improved the quality of the paper. There are only a few minor points that need to be addressed.

Answer to the reviewer: We thank the reviewer for the positive evaluation of our revised manuscript.

1) On page 13, the 11th and 12th lines from the bottom: although the authors stated that ILC2s from CRC patients' PBMCs produced more IL-13 and IL-5 compared to ILC2s from HDs' PBMCs, there is no significant difference for IL-5 (Fig. 4e & f). The authors should delete "and IL-5" from the sentence.

Answer to the reviewer: as suggested by the reviewer, we have removed "and IL-5" from the sentence.

2) It is better to add results on the in vivo tumor growth in ILC2 KO mice as a part of Fig. 7a.

Answer to the reviewer: as suggested by the reviewer, we have included the in vivo tumor growth as the new Figure panel 7b.

3) On page 27, the 6th to 8th lines from the bottom, the authors added a new sentence for the tumor-type dependent benefit of targeting ILC2s. A paper (PMID: 32101749) demonstrating the positive correlation between IL-33 expression and survival of melanoma patients should be cited here to show an example.

Answer to the reviewer: as suggested by the reviewer, we have included the reference in the main manuscript text.